# Light-dependent single-cell heterogeneity in the chloroplast redox state regulates cell fate in a marine diatom

Avia Mizrachi[1], Shiri Graff van Creveld[1], Orr H Shapiro[2], Shilo Rosenwasser[1,3], Assaf Vardi[1]*

[1]Department of Plant and Environmental Sciences, Weizmann Institute of Science, Rehovot, Israel; [2]Department of Food Quality and Safety, Institute of Postharvest and Food Sciences, Agricultural Research Organization, The Volcani Center, Rishon LeZion, Israel; [3]The Robert H. Smith Institute of Plant Sciences and Genetics in Agriculture, The Hebrew University of Jerusalem, Rehovot, Israel

**Abstract** Diatoms are photosynthetic microorganisms of great ecological and biogeochemical importance, forming vast blooms in aquatic ecosystems. However, we are still lacking fundamental understanding of how individual cells sense and respond to diverse stress conditions, and what acclimation strategies are employed during bloom dynamics. We investigated cellular responses to environmental stress at the single-cell level using the redox sensor roGFP targeted to various organelles in the diatom *Phaeodactylum tricornutum*. We detected cell-to-cell variability using flow cytometry cell sorting and a microfluidics system for live imaging of oxidation dynamics. Chloroplast-targeted roGFP exhibited a light-dependent, bi-stable oxidation pattern in response to $H_2O_2$ and high light, revealing distinct subpopulations of sensitive oxidized cells and resilient reduced cells. Early oxidation in the chloroplast preceded commitment to cell death, and can be used for sensing stress cues and regulating cell fate. We propose that light-dependent metabolic heterogeneity regulates diatoms' sensitivity to environmental stressors in the ocean.
DOI: https://doi.org/10.7554/eLife.47732.001

*For correspondence:
assaf.vardi@weizmann.ac.il

Competing interests: The authors declare that no competing interests exist.

## Introduction

Diatoms are considered amongst the most successful and diverse eukaryotic phytoplankton groups, and are estimated to contribute 20% of global net primary production (*Armbrust, 2009*; *Nelson et al., 1995*; *Malviya et al., 2016*). They form massive blooms and are thus central to the biogeochemical cycling of important elements such as carbon, nitrogen, phosphorus, iron and silica, in addition to their important role at the base of marine food webs (*Armbrust, 2009*; *Nelson et al., 1995*; *Morel and Price, 2003*; *Strass and Nöthig, 1996*; *Tréguer et al., 2018*). As other phytoplankton, diatoms need to constantly acclimate to physicochemical gradients in a fluctuating environment. They are exposed to stress from different biotic and abiotic origins such as grazing, viruses, bacteria, allelopathic interactions, light availability, and nutrient limitations (*Bidle, 2015*; *Bidle, 2016*; *Ianora et al., 2006*; *Thamatrakoln et al., 2013*; *Thamatrakoln et al., 2012*; *Kimura and Tomaru, 2015*; *van Tol et al., 2017*; *Vardi et al., 2006*). Importantly, induction of programmed cell death (PCD) in response to different stressors has been suggested as an important mechanism contributing to the fast turnover of phytoplankton and the rapid bloom demise (*Bidle, 2015*; *Bidle, 2016*; *Vardi et al., 2007*).

Recent studies suggested that diatoms can differentially respond to diverse environmental cues based on compartmentalized redox fluctuations that also mediate stress-induced PCD (*Graff van Creveld et al., 2015*; *Rosenwasser et al., 2014*; *Volpert et al., 2018*). Reactive oxygen species

**eLife digest** Microscopic algae, such as diatoms, are widely spread throughout the oceans, and are responsible for half of the oxygen we breathe. At certain times of the year these algae grow very rapidly to form large "blooms" that can be detected by satellites in space. These blooms are generally short-lived because the algae are either eaten by other marine organisms, run out of nutrients, or die as a result of being infected by viruses or bacteria. However, some diatom cells survive the end of the bloom and go on to generate new blooms in the future, but it is still not clear how.

As the bloom collapses, diatoms experience many stressful conditions which can cause active molecules known as reactive oxygen species, or ROS for short, to accumulate inside cells. Normally growing cells also produce low amounts of ROS, which regulate various processes that are important for maintaining a cell's health. However, high amounts of ROS can cause damage, which may lead to a cell's death. Now, Mizrachi et al. investigated why some algae survive while others die in response to stressful conditions, focusing on the amount of ROS that accumulates within the diatom *Phaeodactylum tricornutum*.

Laboratory experiments showed that individual cells of *P. tricornutum* respond differently to environmental stress, forming two distinct groups of either sensitive or resilient cells. Sensitive cells accumulated high levels of ROS within a cell compartment known as the chloroplast and eventually died. Whereas resilient cells were able to maintain low levels of ROS in the chloroplast and survived long after the other cells perished. Populations of genetically identical diatom cells also formed distinct groups of sensitive and resilient cells, demonstrating that these two opposing reactions to stress are not caused by genetic differences between cells.

Lastly, Mizrachi et al. showed that how diatoms acclimate to stress depends on the amount of light they are exposed to. When in the dark, all cells became sensitive to oxidative stress, without forming distinct groups. But, when exposed to strong light that mimics the ocean surface, cells formed distinct groups within the population. This suggests that light regulates how susceptible these microscopic algae are to environmental stress.

The different responses within a population may serve as a "bet-hedging" strategy, enabling at least some of the cells to survive unpredicted stressful conditions. The next challenge will be to find out whether algae growing in the oceans also use the same strategy and investigate what impact this has on diatom blooms.

DOI: https://doi.org/10.7554/eLife.47732.002

(ROS) are known to play an important role in sensing stress and additional signals across kingdoms, from bacteria to plants and animals (*Vardi et al., 1999*; *Mittler et al., 2011*; *Suzuki et al., 2012*; *D'Autréaux and Toledano, 2007*; *Dietz et al., 2016*). They are produced as byproducts of oxygen-based metabolism in respiration and photosynthesis, by ROS generating enzymes, and due to various stress conditions (*Graff van Creveld et al., 2015*; *D'Autréaux and Toledano, 2007*; *Sheyn et al., 2016*; *Foyer and Noctor, 2016*; *Luo et al., 2014*; *Vardi et al., 2002*; *Waring et al., 2010*). To maintain redox balance and avoid oxidative damage, cells harbor various ROS scavenging enzymes and small antioxidant molecules that regulate and buffer ROS levels, such as glutathione (GSH), ascorbate and NADPH. ROS can cause rapid post-translational modifications of pre-existing proteins through oxidation, affecting their activity faster than changes in gene expression (*D'Autréaux and Toledano, 2007*). The specificity of the ROS signal is derived from the specific chemical species, its concentration, sub-cellular localization, temporal dynamics, and available downstream ROS-sensitive targets (*Graff van Creveld et al., 2015*; *D'Autréaux and Toledano, 2007*; *Sheyn et al., 2016*; *Noctor and Foyer, 2016*; *Owusu-Ansah and Banerjee, 2009*; *Tsukagoshi et al., 2010*). Therefore, ROS production and redox metabolic networks can be used to sense and integrate information of both the metabolic state of the cell and its microenvironment.

$H_2O_2$ is a relatively mild and stable ROS that can accumulate in cells due to various stress conditions, thus often serves as a signaling molecule (*Vardi et al., 1999*; *Mittler et al., 2011*; *Suzuki et al., 2012*; *D'Autréaux and Toledano, 2007*; *Dietz et al., 2016*; *Exposito-Rodriguez et al., 2017*). It has a preferential activity towards cysteine residues, and can remodel the

redox-sensitive proteome network (*Rosenwasser et al., 2014*; *Dietz et al., 2016*; *Noctor and Foyer, 2016*). In addition, $H_2O_2$ can diffuse across membranes (depending on membrane properties) and through aquaporins channels (*Bienert and Chaumont, 2014*). Combined properties as lower toxicity, diffusibility and selective reactivity make $H_2O_2$ suitable for studying signaling in various biological systems (*Dietz et al., 2016*; *Noctor and Foyer, 2016*). Since many environmental stressors induce ROS generation (*Vardi et al., 1999*; *Sheyn et al., 2016*; *Waring et al., 2010*; *Exposito-Rodriguez et al., 2017*; *Asada, 2006*; *Diaz et al., 2018*; *Diaz et al., 2013*), application of $H_2O_2$ can reproduce the downstream cellular response (*Graff van Creveld et al., 2015*; *Rosenwasser et al., 2014*). Importantly, $H_2O_2$ application in marine diatoms led to oxidation patterns similar to other environmental stressors such as nutrient limitations, toxic infochemicals and high light, demonstrating that it induces similar physiological responses within the cell (*Figure 1*) (*Graff van Creveld et al., 2015*; *Rosenwasser et al., 2014*). Although the $H_2O_2$ concentrations used in these studies were higher than measured in bulk seawater, which is typically in the nanomolar range (*Zinser, 2018*), local concentrations at the microenvironment of phytoplankton can be significantly higher due to patchiness in time and space (*Stocker, 2012*; *Chrachri et al., 2018*; *Seymour et al., 2017*). Local and temporal production of $H_2O_2$ by the cell itself or by its neighbors could lead to intracellular concentrations which are orders of magnitude higher than measured in bulk seawater in the field, especially during dense blooms or in aggregates and biofilms (*Diaz et al., 2018*; *Diaz et al., 2013*; *Stocker, 2012*; *Chrachri et al., 2018*; *Seymour et al., 2017*). $H_2O_2$ application in the model diatom

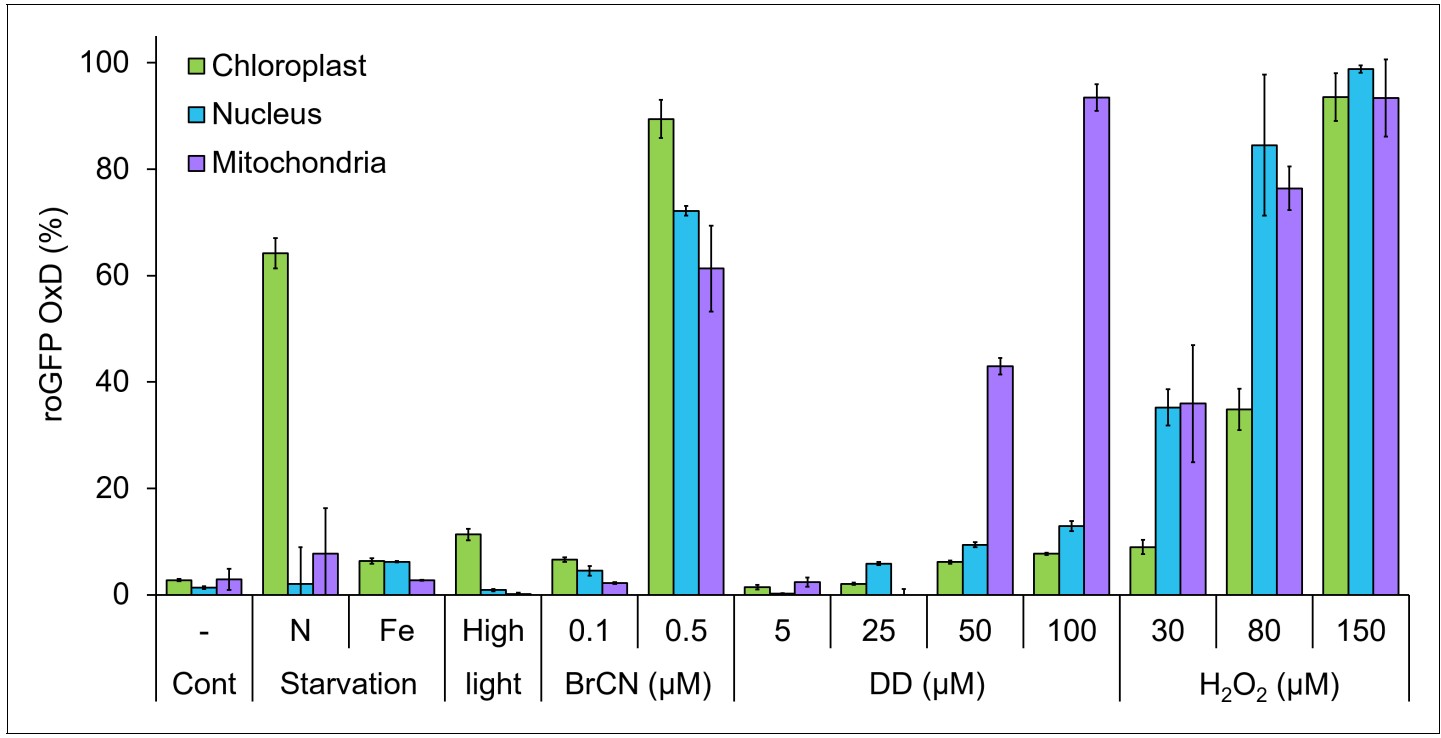

**Figure 1.** Organelle-specific oxidation in response to environmental stressors in *P. tricornutum*. Maximal roGFP degree of oxidation (OxD) measured in the chloroplast, nucleus and mitochondria in response to various environmental stress conditions based on bulk measurements using flow cytometry, data from previously published works. Cont – control untreated cells. N starvation - nitrate limitation 56 hr (See *Rosenwasser et al., 2014*). Fe starvation - iron limitation, day 3 (see *Graff van Creveld et al., 2015*; *Graff van Creveld et al., 2016*). High light, 700 $\mu$mol photons$\cdot$m$^{-2}\cdot$sec$^{-1}$, 38 min (see *Graff van Creveld et al., 2015*). BrCN - cyanogen bromide, 0.1 and 0.5 $\mu$M, 10 min (see *Graff van Creveld et al., 2015*). DD - 2E,4E/Z-decadienal (diatom-derived unsaturated aldehyde), 5, 25, 50 and 100 $\mu$M, 3 hr (see *Graff van Creveld et al., 2015*). $H_2O_2$ – 30, 80, and 150 $\mu$M, ~30 min (see *Graff van Creveld et al., 2015*). Bars represent average of biological triplicates, error bars are standard error of the mean.
DOI: https://doi.org/10.7554/eLife.47732.003

The following source data is available for figure 1:

**Source data 1.** Organelle-specific oxidation in response to environmental stressors in *P. tricornutum*.
DOI: https://doi.org/10.7554/eLife.47732.004

*P. tricornutum* also led to the induction of cell death, in a dose-dependent manner, with characteristics of PCD that included externalization of phosphatidylserine, DNA laddering, and compromised cell membrane (*Graff van Creveld et al., 2015*). Moreover, early oxidation of the mitochondrial GSH pool preceded subsequent cell death at the population level following exposure to $H_2O_2$ or diatom-derived infochemicals (*Graff van Creveld et al., 2015*; *Graff van Creveld et al., 2016*).

In the current work, we investigated phenotypic variability within diatom populations in response to oxidative stress. Classically, phenotypic variability has been studied primarily in established bacterial, yeast and mammalian model systems with little ecological relevance (*Balaban, 2011*; *Raj and van Oudenaarden, 2008*). However, examination of phenotypic heterogeneity in phytoplankton cells in response to stress is still underexplored, and research has been carried out primarily at the population level, masking heterogeneity. Cell-to-cell variability could result in different cellular strategies employed by the population to cope with environmental stress, and may provide important insights into cell survival during bloom succession. We established single-cell approaches to measure in vivo ROS dynamics in the model diatom *P. tricornutum* using flow cytometry and microfluidics live-imaging microscopy. We used *P. tricornutum* strains expressing redox-sensitive GFP (roGFP) targeted to different sub-cellular compartments (*Rosenwasser et al., 2014*) and exposed cells to oxidative stress and high light conditions. The oxidation of roGFP is reversible, and can be quantified using ratiometric fluorescence measurements (*Meyer, 2008*). The roGFP oxidation degree (OxD) reports the redox potential of the GSH pool ($E_{GSH}$), which represents the balance between GSH and its oxidized form (*Meyer, 2008*). Therefore, roGFP OxD provides an important metabolic readout of the redox state of the cell, and represents the oxidation state of native proteins in the monitored organelles (*Rosenwasser et al., 2014*; *Meyer, 2008*). We uncovered a previously uncharacterized phenotypic heterogeneity in the response of a marine diatom to oxidative stress and high light. Furthermore, we revealed a specific link between oxidation patterns in the chloroplast and cell fate regulation.

## Results

### Bi-stable chloroplast roGFP oxidation in response to oxidative stress reveals distinct subpopulations

Previous works demonstrated that *P. tricornutum* exhibits organelle-specific oxidation patterns in responses to diverse environmental stress conditions (*Figure 1*) (*Graff van Creveld et al., 2015*; *Rosenwasser et al., 2014*; *Graff van Creveld et al., 2016*). However, these findings were based on bulk measurements averaging the phenotypes within the population. In order to investigate heterogeneity within the population, we measured the response to oxidative stress in *P. tricornutum* strains expressing roGFP targeted to the chloroplast, nucleus and mitochondria at single-cell resolution using flow cytometry. At steady-state conditions without perturbations, roGFP OxD distribution in the population had a single distinct peak, representing a reduced state at all examined compartments (*Figure 2A,E,I* and *Figure 2—figure supplements 1–2*). Next, we examined the response of cells to oxidative stress by treatment with $H_2O_2$ at concentrations that led to oxidation patterns similar to other environmental stressors (*Figure 1*), and that also led to death of part of the population (*Graff van Creveld et al., 2015*).

Chloroplast-targeted roGFP (chl-roGFP) exhibited a distinct bimodal distribution following treatments of 50–100 µM $H_2O_2$, revealing two distinct subpopulations of 'oxidized' and 'reduced' cells (*Figure 2B–D* and *Figure 2—figure supplement 3A–C*). These subpopulations emerged within the first few minutes after $H_2O_2$ addition (*Figure 2B–D*). The existence of these subpopulation is masked in bulk analysis (*Figure 2—figure supplement 1*), demonstrating the importance of single-cell measurements. In the 'oxidized' subpopulation, roGFP completely oxidized (~100%) in response to $H_2O_2$, reaching a similar distribution of the fully oxidized positive control (200 µM $H_2O_2$) (*Figure 2B–D* and *Figure 2—figure supplement 3G*). However, in the 'reduced' subpopulation roGFP reached lower values of 30–43% OxD within 2–24 min post treatments, and then gradually recovered (*Figure 2B–D* and *Figure 2—figure supplement 3G*). Only a minor fraction of the cells displayed intermediate oxidation, suggesting that these subpopulations represent discrete redox states. Interestingly, a larger fraction of cells was within the 'oxidized' subpopulation at 20–25 min post treatment compared to later time points, indicating that some cells were able to recover during this time (*Figure 2M*). The

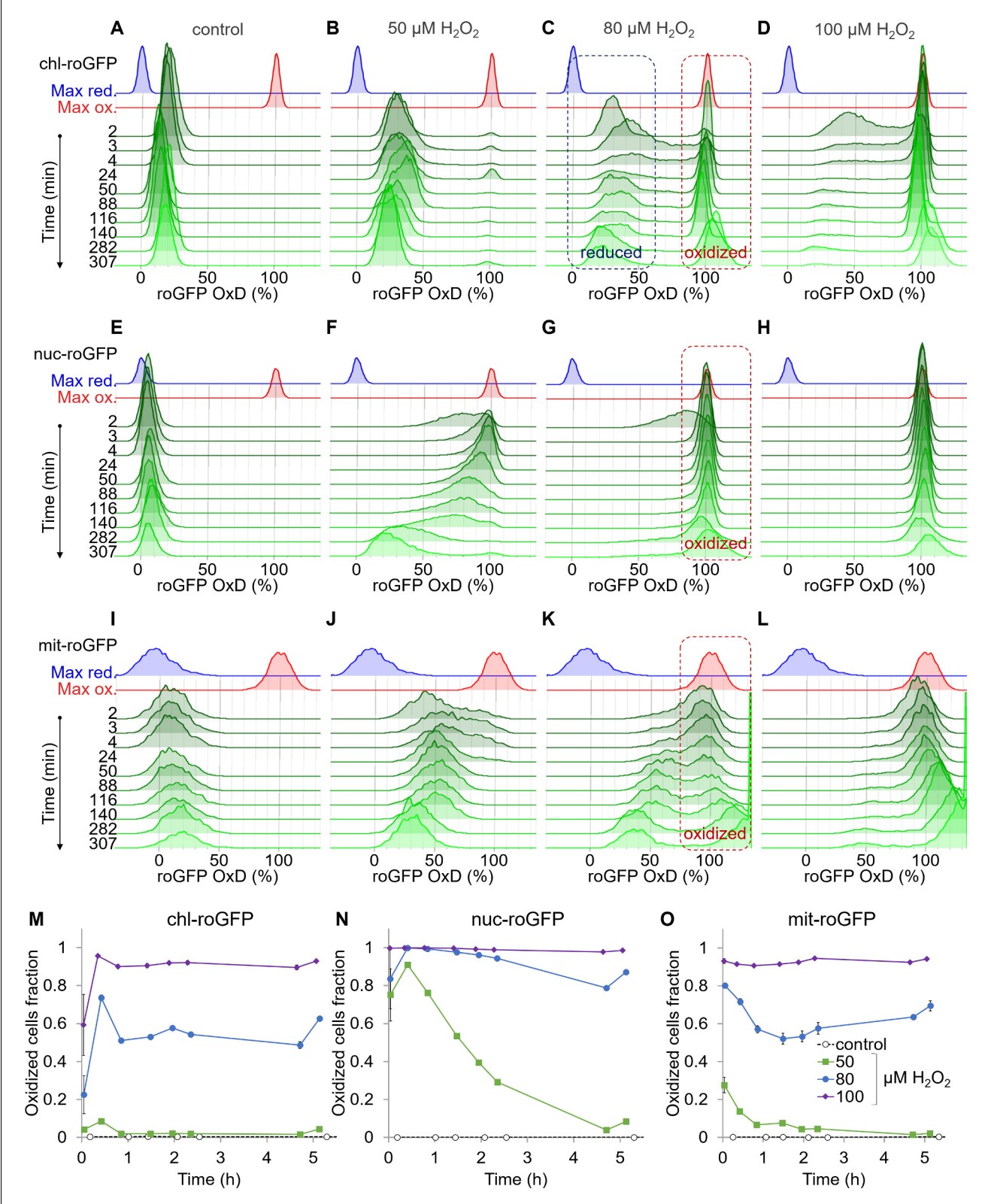

**Figure 2.** Organelle-specific oxidation of roGFP in response to $H_2O_2$ reveals heterogeneity at the single-cell level. The distribution of roGFP OxD in the population over time was measured by flow cytometry in *P. tricornutum* cells expressing roGFP targeted to the chloroplast (chl-roGFP, (A–D, M)), nucleus (nuc-roGFP, (E–H, N)) and mitochondria (mit-roGFP, (I–L, O)). (A–L) Oxidation of roGFP in response to 0 μM (A,E,I), 50 μM (B,F,J), 80 μM (C,G, K), and 100 μM (D,H,L) $H_2O_2$. Maximum reduction (blue) and oxidation (red) of roGFP following additions of 2 mM Dithiothreitol (DTT) or 200 μM $H_2O_2$

*Figure 2 continued on next page*

eLIFE Research article | Ecology | Microbiology and Infectious Disease

*Figure 2 continued*

respectively are shown as a reference. The 'oxidized' and 'reduced' subpopulations are marked by red and blue dashed boxes respectively (**C, G, K**). The experiment was done in triplicates, for visualization the first replica is shown except for the first 4 min in which all replicates are shown for higher temporal resolution. Each histogram consists of >8000 (**A–D**), >5900 (**E–H**) and >1400 (**I–L**) roGFP-positive cells. Measurements of >100% OxD may result from increased auto-fluorescence leakage after long exposure to stress (*Figure 2—figure supplements 5–6*). (**M–O**) The fraction of the 'oxidized' subpopulation over time upon exposure to 0–100 µM $H_2O_2$. Mean ± SEM, n = 3 biological repeats. SEM lower than 0.018 are not shown.
DOI: https://doi.org/10.7554/eLife.47732.005

The following source data and figure supplements are available for figure 2:

**Source data 1.** Flow cytometry measurements of organelle-specific oxidation of roGFP in response to $H_2O_2$ over time.
DOI: https://doi.org/10.7554/eLife.47732.012

**Figure supplement 1.** Organelle specific roGFP mean oxidation in response to $H_2O_2$ application in *P. tricornutum*.
DOI: https://doi.org/10.7554/eLife.47732.006

**Figure supplement 2.** Distribution of roGFP oxidation in different subcellular compartments of *P. tricornutum* cells at steady state.
DOI: https://doi.org/10.7554/eLife.47732.007

**Figure supplement 3.** Oxidation of chl-roGFP reveals distinct subpopulations in response to $H_2O_2$ treatment.
DOI: https://doi.org/10.7554/eLife.47732.008

**Figure supplement 4.** Heterogeneity in mit-roGFP response to $H_2O_2$.
DOI: https://doi.org/10.7554/eLife.47732.009

**Figure supplement 5.** Auto-fluorescence leakage into i405 channel in roGFP strains following $H_2O_2$ treatment.
DOI: https://doi.org/10.7554/eLife.47732.010

**Figure supplement 6.** Auto-fluorescence leakage into i488 channel in roGFP strains following $H_2O_2$ treatment.
DOI: https://doi.org/10.7554/eLife.47732.011

proportion between these subpopulations stabilized after 46–51 min post treatment, and was $H_2O_2$-dose dependent, as more cells were within the 'oxidized' subpopulation at higher $H_2O_2$ concentrations (*Figure 2M*). The quick emergence of stable co-existing 'oxidized' and 'reduced' subpopulations exposed underlying heterogeneity within the diatom population, resulting in a differential response to oxidative stress.

This clear bi-stable pattern was unique to the chloroplast-targeted roGFP. Nuclear-targeted roGFP displayed a continuous distribution in response to $H_2O_2$ treatments, and no distinct subpopulations could be observed (*Figure 2F–H*). Within minutes post treatment, nuclear roGFP exhibited fast oxidation even in response to a low $H_2O_2$ concentration of 50 µM, which had only a mild effect on the chloroplast (*Figure 2B,F*). At that concentration, nuclear roGFP oxidation was followed by a gradual and much slower recovery, which lasted >5 hr post treatment (*Figure 2F*). At higher concentrations, the entire population was oxidized within 3 min post treatment, and most cells remained stably oxidized >5 hr post treatment (*Figure 2G–H*). The mitochondria-targeted roGFP exhibited a heterogeneous redox response, as seen in the 80 µM and 100 µM $H_2O_2$ treatments starting at ~24 min post treatment (*Figure 2K–L*). However, distinct subpopulations were not clearly separated until later stages, and were not detected consistently in different experiments (*Figure 2K–L* and *Figure 2—figure supplement 4*). Therefore, we chose to focus on the chl-roGFP strain, which revealed two discrete subpopulations.

## Oxidation of chl-roGFP precedes the induction of cell death

Next, we examined the possible link between early chloroplast $E_{GSH}$ oxidation and subsequent $H_2O_2$-dependent cell death. We quantified cell death 24 hr post $H_2O_2$ treatment using flow cytometry measurements of Sytox green staining, which selectively stains nuclei of dead and dying cells. The fraction of 'oxidized' cells 1–2 hr post treatment was correlated with the fraction of dead cells at 24 hr (*Figure 3A*, $R^2 = 0.89$, $p=2.2\cdot10^{-16}$), suggesting that early oxidation in the chloroplast in distinct subpopulations may predict cell fate at much later stages.

To investigate directly the link between early chl-roGFP oxidation and subsequent cell death, we used fluorescence-activated cell sorting (FACS) to sort cells based on chl-roGFP oxidation and measured their survival. Single cells of the 'oxidized' and 'reduced' subpopulations were sorted into fresh media at different time-points following the addition of 80 µM $H_2O_2$, and colony forming units (CFU) were counted to assess survival 3.5–9 weeks later (*Figure 3B* and *Figure 3—figure supplement 1*). The CFU assay provides a direct link between chl-roGFP oxidation and the ability of

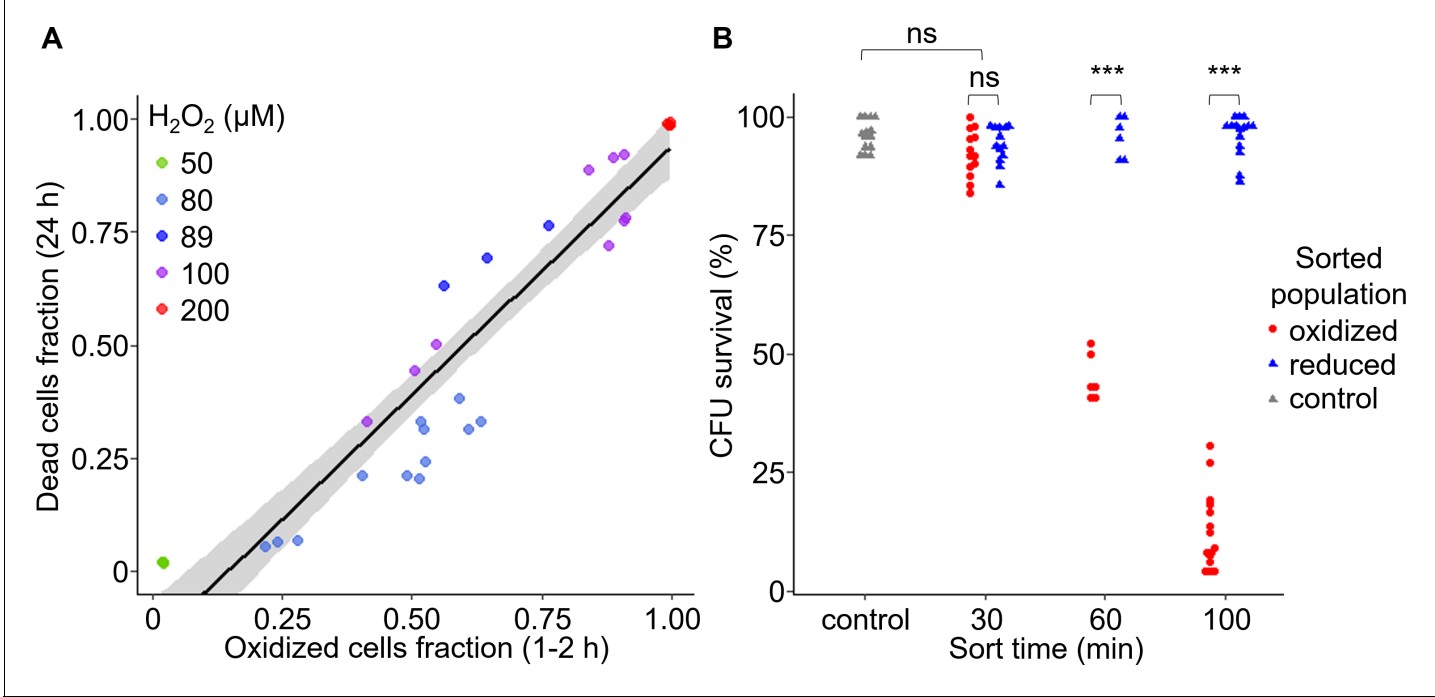

**Figure 3.** Early oxidation of chl-roGFP in a subpopulation precedes the induction of cell death. (A) The fraction of dead cells measured by Sytox positive staining 24 hr post 50–200 $H_2O_2$ treatment as a function of the fraction of chl-roGFP 'oxidized' cells 1–2 hr post treatment (see *Figure 2C*). N ≥ 3 biological repeats per treatment, individual samples are shown, data of 5 independent experiments. Linear model is shown with 95% confidence interval in gray, formula: *y = 1.09x-0.16*, $R^2$ = 0.89. (B) Colony forming units (CFU) survival of individual cells that were sorted into fresh media for regrowth based on their chl-roGFP oxidation at different times following treatment with 80 µM $H_2O_2$. Control – untreated chl-roGFP positive cells that were sorted regardless of degree of oxidation. CFU % survival was measured by the number of CFU divided by the number of sorted cells. N ≥ 6 biological repeats, individual repeats are shown, each of 24–48 separately sorted single cells. Oxidized – red circles; reduced – blue triangles; control – gray triangles. *** – p<0.001; ns – non-significant.
DOI: https://doi.org/10.7554/eLife.47732.013

The following source data and figure supplements are available for figure 3:

**Source data 1.** Early chl-roGFP oxidation and subsequent cell death in response to $H_2O_2$.
DOI: https://doi.org/10.7554/eLife.47732.016
**Source data 2.** Colony forming units single-cell survival of subpopulations following $H_2O_2$ treatment and sorting.
DOI: https://doi.org/10.7554/eLife.47732.017
**Figure supplement 1.** Schematic representation of sorting experiments layout and post-sorting analyses.
DOI: https://doi.org/10.7554/eLife.47732.014
**Figure supplement 2.** Mortality of sorted subpopulations following $H_2O_2$ treatment.
DOI: https://doi.org/10.7554/eLife.47732.015

individual cells to proliferate, and in addition enables to generate clonal populations which were also used for downstream analyses (*Figure 3—figure supplement 1*). When sorted 30 min post treatment, the 'oxidized' subpopulation exhibited a high survival rate (92.3 ± 1.4%) that was similar to the 'reduced' subpopulation (94.1 ± 1.1%, p=0.24, paired t-test) and to sorted untreated control (96 ± 0.9%, p=0.29, Dunnett test, *Figure 3B*). However, at later time-points the survival of the 'oxidized' subpopulation gradually diminished, and was significantly lower than both the 'reduced' and control sorted cells (p<0.001 for all comparisons, paired t-test for comparisons with 'reduced' cells, Dunnett test for comparisons with control). When sorted 60 min post treatment almost half of the 'oxidized' cells recovered (45.1 ± 2%), but when sorted 100 min following treatment only 12.7 ± 2.1% survived (p<0.001, Tukey test, *Figure 3B*). These results suggest that after a distinct exposure time, cell death is induced in an irreversible manner in the 'oxidized' subpopulation. In contrast, the 'reduced' subpopulation from the same culture and treatment exhibited a high survival rate similar to the control at all time-points examined, demonstrating its resilience to the stress (*Figure 3B*, P≥0.86 for all comparisons, Dunnett test). Corroborating these findings, cell death measurements

using Sytox staining of sorted subpopulations also showed higher mortality in the 'oxidized' cells compared to the 'reduced' and control cells, which remained viable (*Figure 3—figure supplement 2*). Taken together, these results demonstrate that the 'oxidized' subpopulation was sensitive to the oxidative stress which led to induction of cell death in those cells, while the 'reduced' subpopulation was able to survive. In addition, we detected a distinct phase of 'pre-commitment' to cell death, ranging approximately 30–60 min in most cells, during which despite the strong oxidation in the chloroplast the fate of the 'oxidized' subpopulation is still reversible upon removal of the stress, and they are still able to survive. After this initial phase, the 'oxidized' cells were not able to survive even when the stress was removed from the system by sorting into fresh media.

## Early oxidation of chloroplast $E_{GSH}$ is linked to cell fate determination at the single-cell level

In order to track oxidation dynamics and subsequent cell fate of individual cells, we established a microfluidics platform for in vivo long-term epifluorescence imaging, under controlled flow, light and temperature conditions customized specifically for diatom cells (*Figure 4—figure supplement 1* and *Videos 1–2*). We introduced cells expressing chl-roGFP into a custom-made microfluidics device, let the cells settle, and introduced treatments of either 80 µM $H_2O_2$ or fresh media (control) continuously for 2.5–3 hr, after which the treatment was washed by fresh media (see Materials and methods, *Figure 4—figure supplement 1A*). In addition, the use of microfluidics enabled imaging of the basal OxD state of single cells prior to treatment, as well as the introduction of Sytox green at the end of the experiment to visualize cell death. We detected the distinct 'oxidized' and 'reduced' subpopulations following 80 µM $H_2O_2$ treatment, similar to the flow cytometry experiments (*Figure 4C,F*, and *Video 1*). However, no clear differences were observed in their OxD prior to treatment (*Figure 4—figure supplement 2* and *Video 1*). The separation between the subpopulations emerged within 20 min of exposure to 80 µM $H_2O_2$, and remained stable over the course of the experiment with the 'oxidized' subpopulation maintaining a high OxD above 80% (*Figure 4F*, *Figure 4—figure supplement 2B* and *Video 1*). The 'reduced' subpopulation exhibited an immediate response to $H_2O_2$ comparable with flow cytometry measurements, from 25–45% OxD before treatment to 30–65% OxD during the first 20 min post 80 µM $H_2O_2$ treatment (*Figure 4F*, *Figure 4—figure supplement 2B*, and *Video 1*). Following this initial oxidation, the 'reduced' cells recovered gradually over the next hours, reducing to 5–25% OxD 8 hr post treatment, even below the initial basal state (*Figure 4F*, *Video 1*). A gradual slow reduction was also observed in control cells over the course of the experiment (*Figure 4E*, *Video 2*), which may represent acclimation to the experimental setup or a diurnal redox alteration. Control cells did not oxidize in response to addition of fresh media (*Figure 4E* and *Figure 4—figure supplement 2A*), excluding the possibility that the oxidation observed in 80 µM $H_2O_2$ treated cells was due to shear stress during treatment.

We detected a clear correlation between initial oxidation in the chloroplast in response to oxidative stress and subsequent cell fate (*Figure 4A–G*). Cells that exhibited high chl-roGFP oxidation within the first 40 min also died at a much later stage, while cells that maintained a lower OxD were able to recover (*Figure 4G*). In addition, cells of the 'reduced' subpopulation and of control treatment were able to proliferate, further demonstrating their viability under these conditions (*Videos 1–2*). Logistic regression modeling of cell death as a function of chl-roGFP OxD at this time-point revealed a

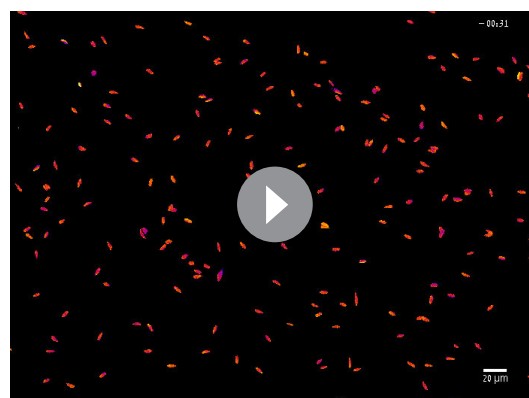

**Video 1.** Microfluidics in vivo imaging of chl-roGFP oxidation over time following $H_2O_2$ treatment. Oxidation of chl-roGFP cells was imaged over time using a customized microfluidic setup and epifluorescence microscopy with controlled flow, light and temperature conditions (see Materials and methods). Movie of chl-roGFP OxD in pseudo-color of cells treated with 80 µM $H_2O_2$ at time 0, the first frame is before treatment. Time stamp represents time post treatment (hh:mm). Color bar as in *Figure 4*.
DOI: https://doi.org/10.7554/eLife.47732.024

threshold of ~74% OxD, which discriminated with high accuracy (98.8%) between cells that subsequently died and cells that survived the treatment (0.8% false positive, 1.7% false negative; *Figure 4G* and *Figure 4—figure supplement 3*). These results corroborate the flow-cytometry analysis, and demonstrate that under these conditions early chloroplast $E_{GSH}$ response is linked to subsequent cell fate determination at the single-cell level.

## The distinct subpopulations derive from phenotypic variability and not from variable genetic backgrounds

The differential chloroplast oxidation of the observed subpopulations could be due to genetic variability or due to phenotypic plasticity within the population. To differentiate between the two scenarios, we sorted chl-roGFP individual cells of the 'oxidized' and 'reduced' subpopulations 30 and 100 min post 80 μM $H_2O_2$ treatment as well as untreated control cells, and regrew them to generate clonal populations derived from cells exhibiting specific phenotypes. The clonal progeny cultures were subsequently exposed to 80 μM $H_2O_2$ and their chl-roGFP oxidation was measured (*Figure 3—figure supplement 1*). The two distinct subpopulations were detected in all the clones measured, and the fraction of the 'oxidized' subpopulation was again correlated with cell death (*Figure 5* and *Figure 5—figure supplement 1*). Therefore, the different subpopulations observed did not originate from genetic differences, but rather represent phenotypic variability within isogenic populations.

## The 'oxidized' subpopulation is enriched with cells at $G_1$ phase

One possible source for phenotypic variability in genetically homogenous populations can be explained by differences in the cell cycle phase, as the cell cycle is linked to metabolic changes including redox oscilations (*Papagiannakis et al., 2017*; *Burhans and Heintz, 2009*; *Mathis and Ackermann, 2016*; *Diaz-Vivancos et al., 2015*). Therefore, we sorted the 'oxidized' and 'reduced' subpopulations 30 min following 80 μM $H_2O_2$ treatment into a fixation solution, and stained the fixed cells with 4′,6-diamidino-2-phenylindole (DAPI) to quantify DNA content for cell cycle analysis. The sorted 'oxidized' subpopulation had a higher fraction of cells at $G_1$ (86.9 ± 1.8%) compared to control untreated cells (76.8 ± 0.7%, p=0.0024, Tukey test, *Figure 6*). The 'reduced' subpopulation on the other hand had a smaller fraction of $G_1$ cells (68.7 ± 2.2%) compared to both control (p=0.011, Tukey test) and 'oxidized' cells (p=0.0001, paired t-test), and exhibited a larger fraction of $G_2$/M cells (*Figure 6*). These results demonstrate that although cell cycle phase alone cannot explain the differences between the subpopulations, it is linked to the chloroplast $E_{GSH}$ response to oxidative stress and may represent an important factor that affects $H_2O_2$ sensitivity in the population.

## The bimodal chloroplast redox response is light dependent

Photosynthesis is a major source for reductive power as well as ROS in algal cells, and exposure to dark was shown to increase sensitivity to oxidative stress in another marine diatom (*Volpert et al., 2018*). Therefore, we hypothesized that light regime will affect the bimodal pattern of chl-roGFP following oxidative stress, and investigated the effects of short exposure to darkness during daytime. Cells were treated with 0–100 μM $H_2O_2$ and were immediately moved to the dark for 90 min, after which they were moved back to the light (dark treated, *Figure 7A–C* and *Figure 7—figure supplement 1*). These cells were compared to cells that were kept in the light during this time (light treated). The transition to the dark caused an immediate oxidation of the basal chl-roGFP OxD (without $H_2O_2$ treatment), reaching a peak within 15 min (*Figure 7A,D*). Then, while still under dark, chl-roGFP gradually reduced while maintaining a continuous distribution (*Figure 7A,D*). Upon shifting back to the light, chl-roGFP reduced within 2 min back to its basal state prior to dark exposure (*Figure 7A,D*). The dark mediated oxidation was specific to the chloroplast and was not detected in the nucleus (*Figure 7—figure supplement 2*), demonstrating organelle specificity of these redox fluctuations.

The transition to the dark eliminated the bimodal pattern of chl-roGFP oxidation in response to $H_2O_2$ (*Figure 7C*). No distinct subpopulations were observed while cells were under darkness even in cells treated with low $H_2O_2$ doses (*Figure 7B* and *Figure 7—figure supplement 1B–C*). The transition to the dark increased $H_2O_2$ sensitivity in the entire population, and following treatment of 80 μM $H_2O_2$ and transition to the dark chl-roGFP fully oxidized in the entire population and remained

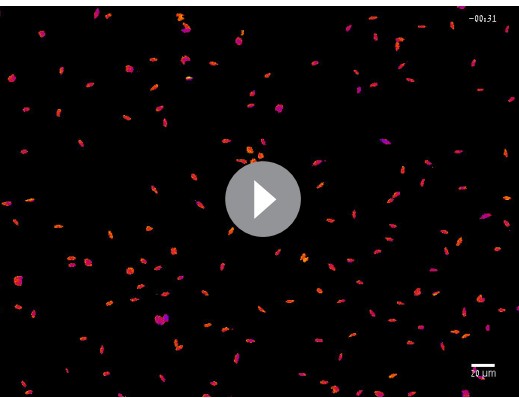

**Video 2.** Microfluidics in vivo imaging of basal chl-roGFP oxidation over time. Oxidation of chl-roGFP cells was imaged over time using a customized microfluidic setup and epifluorescence microscopy with controlled flow, light and temperature conditions (see Materials and methods). Movie of chl-roGFP OxD in pseudo-color of cells treated with fresh media (control) at time 0, the first frame is before treatment. Time stamp represents time post treatment (hh:mm). Color bar as in **Figure 4**.
DOI: https://doi.org/10.7554/eLife.47732.025

stably oxidized even after transition back to the light (**Figure 7C**). The bimodal pattern was regained only upon transition back to the light, and only at lower doses of 30 µM and 50 µM $H_2O_2$, in which some or most cells were able to recover following this transition (**Figure 7B** and **Figure 7—figure supplement 1C**). In accordance with the higher chl-roGFP oxidation, 'dark' treated cultures also exhibited higher mortality compared to 'light' treated cells (p≤0.0064 for all pairs in 50–100 µM $H_2O_2$ treatments, t-test, **Figure 7E**). Therefore, we conclude that the mechanism generating the bimodal response in the chloroplast is light dependent and plays an important role in cell fate regulation in diatoms.

## High light induces variability in chloroplast ROS accumulation and subsequent cell survival

Next, we investigated the effects of high light (HL), a key environmental stressor in the marine ecosystem, which was shown to induce ROS generation in the chloroplast (**Waring et al., 2010**; **Exposito-Rodriguez et al., 2017**; **Asada, 2006**). We hypothesized that HL may cause $H_2O_2$ accumulation in the chloroplast, leading to a similar phenotypic variability as detected in response to $H_2O_2$ treatment. Cells were exposed to HL treatment of 2,000 µmol photons $m^{-2}$ $s^{-1}$ in order to evaluate the effect of HL exposure equivalent to full sunlight in nature (**Long et al., 1994**), and as was used in previous studies in *P. tricornutum* (**Lepetit et al., 2013**). Indeed, 1.5 hr after exposure to HL an 'oxidized' subpopulation started to emerge (**Figure 8A** and **Figure 8—figure supplement 1B**). The fraction of 'oxidized' cells gradually increased over time of HL exposure (**Figure 8A–B** and **Figure 8—figure supplement 1A–F**). These subpopulations were not detected in control cells that were kept under low light (LL, **Figure 8A** and **Figure 8—figure supplement 1G–H**), nor in cells expressing nuclear targeted roGFP that were exposed to HL (**Figure 8—figure supplement 2A–B**), demonstrating the specificity of the redox signal to the chloroplast.

To measure the survival of the subpopulations that emerged under HL conditions, individual cells from the 'oxidized' and 'reduced' subpopulations were FACS sorted after different HL exposure times into agar plates for the single-cell CFU survival assay (see Materials and methods, **Figure 8C–D**). CFU survival of the 'oxidized' subpopulation gradually decreased over time of HL exposure and was significantly lower than both control and 'reduced' cells (p<0.001 for all comparisons. Tukey test was used for comparison with control, ANCOVA was used for comparison with 'reduced' and for the interaction with exposure time, **Figure 8D**). While most 'oxidized' cells survived ≤3 hr HL exposure (77.3 ± 4.2% CFU survival), longer exposure of >6 hr to HL led to only 3.1 ± 0.9% CFU survival (**Figure 8D**). In contrast, the 'reduced' subpopulation in the same HL treatment exhibited high CFU survival of 92.7 ± 3.6%, and maintained high CFU survival at all time-points examined similar to the control (p=0.93, Tukey test, **Figure 8D**). Interestingly, when cells were exposed to 6.3 hr HL and then moved to 1 hr LL the separation between the subpopulations became clearer, with almost no cells with intermediate oxidation states (**Figure 8A** and **Figure 8—figure supplement 1E–F**), resembling the response to $H_2O_2$ treatment (**Figure 2—figure supplement 3B**). To conclude, these findings demonstrate that HL can generate heterogeneity in ROS accumulation in the chloroplast within diatom populations, leading to differences in survival probability and likely affecting sensitivity to additional stressors in the marine environment.

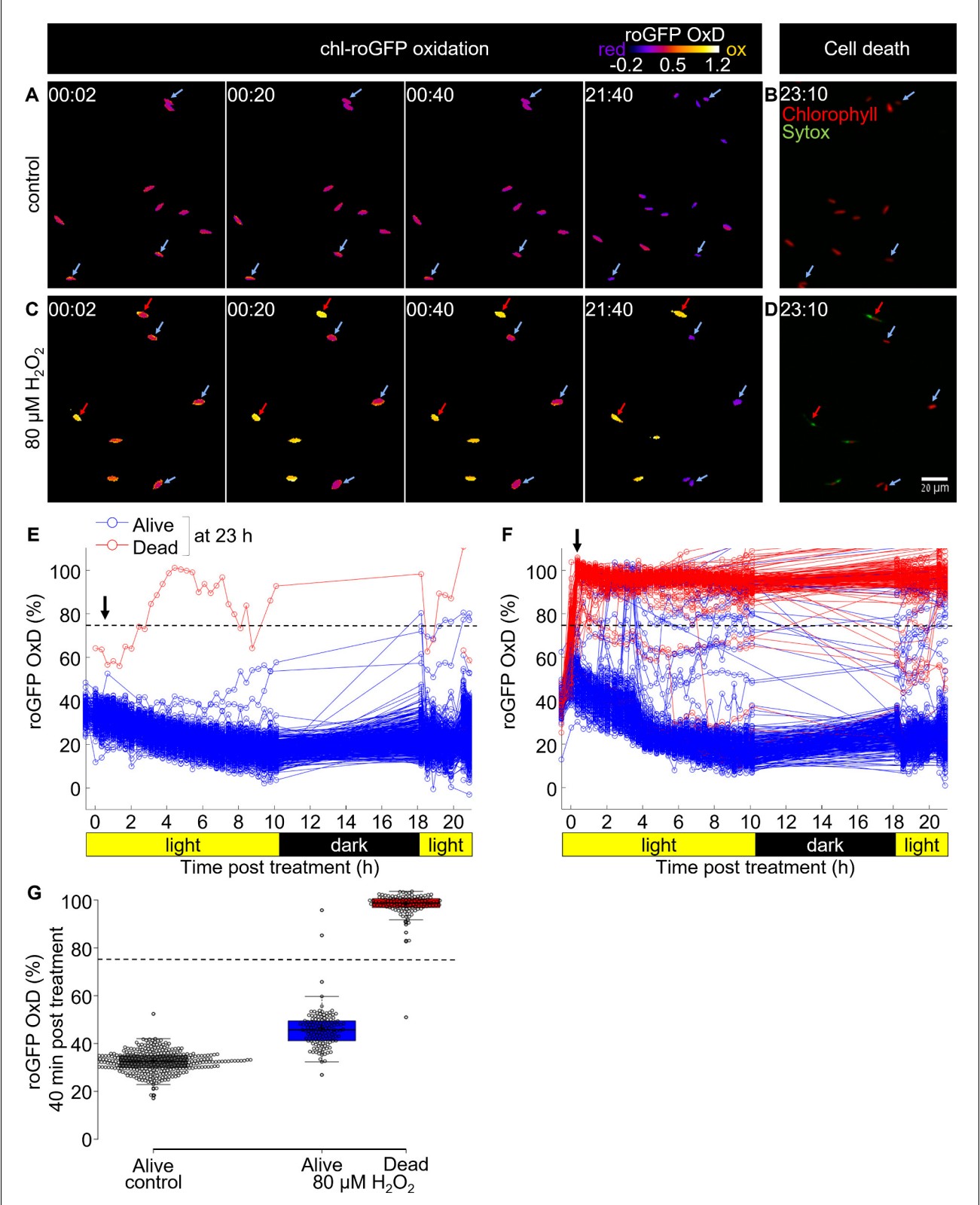

**Figure 4.** Tracking redox dynamics of individual cells using long term in vivo imaging in a microfluidics setup. Oxidation of chl-roGFP was imaged over time using a customized microfluidic setup and epifluorescence microscopy. Cells were imaged following treatment with either fresh media (control; **A, B, E**) or 80 μM $H_2O_2$ (**C, D, F**). To quantify cell death, cells were stained with Sytox ~23 hr post treatment. (**A, C**) Representative frames depicted in pseudo-color of calculated roGFP OxD at different times post treatment (hh:mm). Two subpopulations of 'oxidized' (red arrows) and 'reduced' (blue

*Figure 4 continued on next page*

*Figure 4 continued*

arrows) cells were detected in treated cells. (**B, D**) Overlay of Sytox staining (green, dead cells) and chlorophyll auto-fluorescence (red) at 23:10 hr post treatment. (**E, F**) Quantification of chl-roGFP OxD per cell over time post treatment. Color is based on cell fate as measured ~23 hr post treatment: blue – alive (Sytox negative), red – dead (Sytox positive). OxD values > 110% likely result from auto-fluorescence leakage, and are not shown. (**G**) Box-plot of chl-roGFP OxD 40 min post treatment (arrow in **E, F**) of cells that were grouped based on their cell fate as measured ~23 hr post treatment. Gray – control alive; blue – 80 µM $H_2O_2$ alive; red – 80 µM $H_2O_2$ dead. Circles - single cells; cross – mean; colored box – 1st to 3rd quartiles; horizontal line within the box – median. Individual cells are shown. (**E–G**) N ≥ 250 cells per treatment from ≥3 different fields of view. Horizontal dashed line represents the chl-roGFP oxidation threshold used for discriminating cells that subsequently died or survived.

DOI: https://doi.org/10.7554/eLife.47732.018

The following source data and figure supplements are available for figure 4:

**Source data 1.** Early chl-roGFP OxD and subsequent cell fate of single cells in response to $H_2O_2$ in a microfluidics setup.

DOI: https://doi.org/10.7554/eLife.47732.023

**Figure supplement 1.** Microfluidics experimental layout.

DOI: https://doi.org/10.7554/eLife.47732.019

**Figure supplement 2.** Tracking early chl-roGFP oxidation dynamics in single cells following oxidative stress using in vivo imaging in a microfluidics setup.

DOI: https://doi.org/10.7554/eLife.47732.020

**Figure supplement 3.** Logistic regression analysis of cell fate vs. chl-roGFP oxidation.

DOI: https://doi.org/10.7554/eLife.47732.021

**Figure supplement 4.** Schematic representation of the microfluidics image analysis pipeline.

DOI: https://doi.org/10.7554/eLife.47732.022

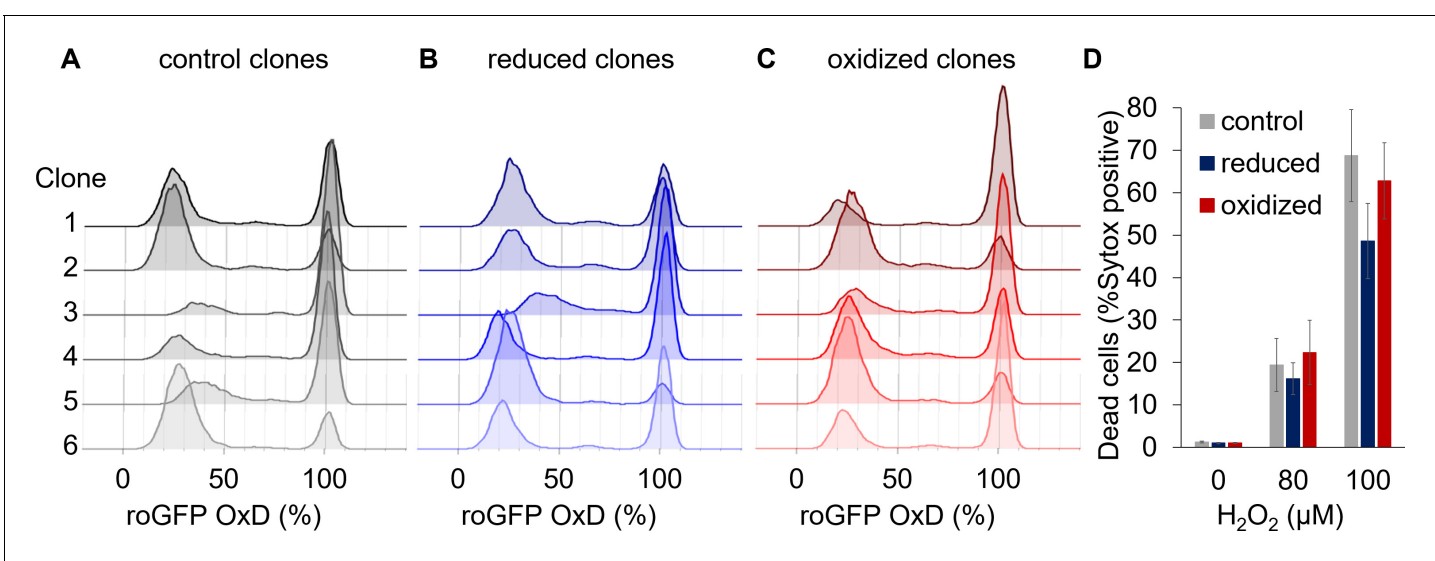

**Figure 5.** Clonal populations derived from sorted cells maintain the bi-stable phenotype of chl-roGFP response to $H_2O_2$. (**A–C**) The distribution of chl-roGFP OxD (%) 40–45 min post 80 µM $H_2O_2$ treatment in clonal populations derived from sorted single cells of different origins, 3 weeks post sorting. The 'reduced' (**B**) and 'oxidized' (**C**) subpopulations were sorted 30 min post 80 µM $H_2O_2$ treatment based on chl-roGFP oxidation (***Figure 3—figure supplement 1***); 'control' (**A**) – clones of untreated cells that were sorted based on positive roGFP fluorescence. Each histogram is of a single clone, ≥9900 cells per histogram, six representative clones per group are shown. (**D**) The fraction of dead cells 24 hr post $H_2O_2$ treatment of the different clones shown in (**A–C**) as measured by positive Sytox staining. Data is shown as mean ± SEM, n = 6 clones per group per treatment.

DOI: https://doi.org/10.7554/eLife.47732.026

The following source data and figure supplement are available for figure 5:

**Source data 1.** The fraction of dead cells post $H_2O_2$ treatment of clones originating from different subpopulations.

DOI: https://doi.org/10.7554/eLife.47732.028

**Figure supplement 1.** Sorted clonal populations maintain the bi-stable phenotype in chl-roGFP response to $H_2O_2$.

DOI: https://doi.org/10.7554/eLife.47732.027

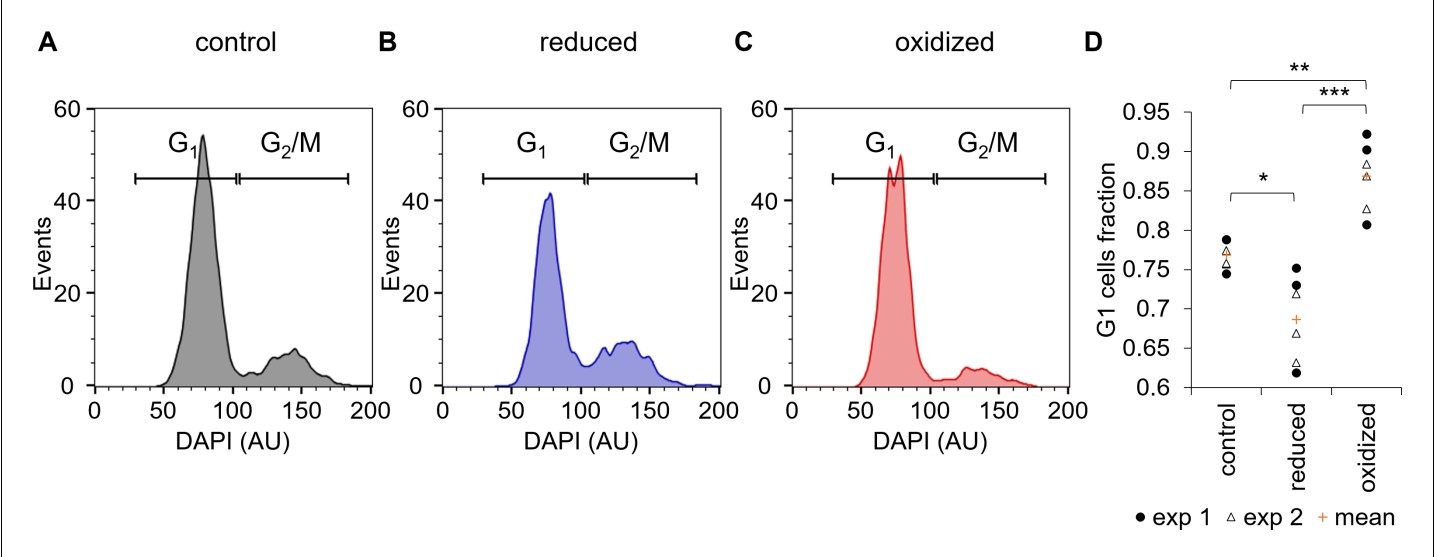

**Figure 6.** Cell cycle analysis of sorted 'reduced' and 'oxidized' subpopulations. (**A–C**) Cell cycle analysis of FACS sorted control untreated cells (**A**) and of 'reduced' (**B**) and 'oxidized' (**C**) subpopulations that were sorted 30 min post 80 µM $H_2O_2$ treatment, based on chl-roGFP OxD. DAPI staining was used for DNA content quantification, gates used to discriminate $G_1$ (two genome copies) and $G_2/M$ (four copies) are marked. (**D**) Fraction of cells within the $G_1$ gate (marked in A-C) in sorted subpopulations. Data points of n = 6 biological repeats of 2 independent experiments (exp 1 and 2 marked with circles and triangles respectively) and the mean (orange plus sign) are shown, 1200–2000 cells analyzed per sample. $P$-values: *=0.011, **=0.0024, ***=0.0001. Tukey test was used for comparisons with control, paired t-test was used for comparing 'reduced' and 'oxidized' subpopulations.

DOI: https://doi.org/10.7554/eLife.47732.029

The following source data is available for figure 6:

**Source data 1.** Cell cycle analysis of sorted subpopulations.

DOI: https://doi.org/10.7554/eLife.47732.030

## Discussion

Our current understanding of the mechanisms that mediate acclimation to environmental stressors in marine microorganisms, including diatoms, has been derived primarily from observations at the population level, neglecting any heterogeneity at the single-cell level. Co-existence of distinct subpopulations that employ diverse cellular strategies can be significant for the survival of this globally important phytoplankton group, as was shown in other microorganisms (*Balaban, 2011*; *Schreiber et al., 2016*; *Sengupta et al., 2017*). Here, we established a novel system for studying phenotypic variability in the marine diatom *P. tricornutum* using flow cytometry and a microfluidics system for live imaging microscopy. Based on a metabolic readout of chloroplast $E_{GSH}$ oxidation, we uniquely identified two distinct subpopulations that emerged as an early response to $H_2O_2$ and high light, demonstrating the importance of phenotypic variability in cell fate regulation in diatoms.

We propose that in diatoms, chloroplast $E_{GSH}$ is involved in sensing specific environmental stress cues and in regulating cell fate (*Figure 9*). The chloroplast is a major source for generation of both ROS and reductive power to generate and recycle NADPH, thioredoxin and GSH (*Dietz et al., 2016*). In plants, chloroplast-generated ROS were demonstrated to be involved in retrograde signaling from the chloroplast and in hypersensitive response cell death (*Dietz et al., 2016*; *Exposito-Rodriguez et al., 2017*; *Van Aken and Van Breusegem, 2015*; *Liu et al., 2007*). In diatoms, specific stress cues can lead to ROS accumulation and $E_{GSH}$ oxidation in the chloroplast, as was shown in response to nitrogen limitation, the diatom-derived toxic infochemicals cyanogen bromide, and HL (*Figure 8A* and *Figure 1*) (*Graff van Creveld et al., 2015*; *Rosenwasser et al., 2014*). Specifically, HL can lead to ROS accumulation in the chloroplast by generation of singlet oxygen ($^1O_2$) in PSII, and by photoreduction of $O_2$ to superoxide ($O_2^-$) in PSI, which in turn is rapidly converted to $H_2O_2$ by superoxide dismutase (SOD) (*Waring et al., 2010*; *Exposito-Rodriguez et al., 2017*; *Asada, 2006*). Redox fluctuations in the chloroplast can serve as a rapid mechanism to perceive specific environmental cues, by regulating key metabolic pathways on the post-translational level.

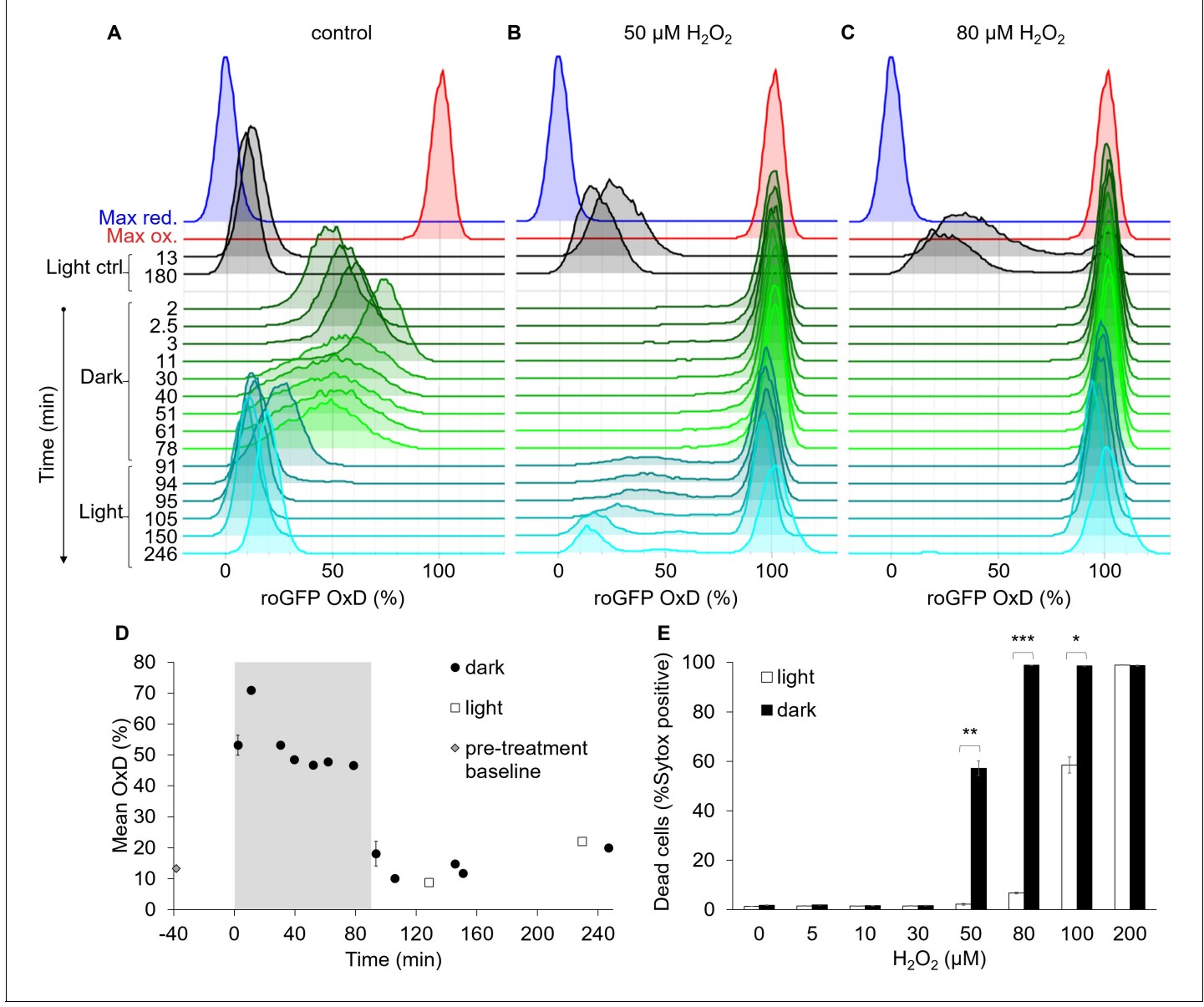

**Figure 7.** The bimodal chl-roGFP oxidation in response to $H_2O_2$ is light-dependent. The effects of a short exposure to darkness during daytime on chl-roGFP oxidation patterns were examined. (A–C) Flow cytometry measurements of chl-roGFP OxD distribution in the population over time. Cells were treated with 0 μM (control, A), 50 μM (B), and 80 μM $H_2O_2$ (C), and were then transitioned to the dark at time 0 (within 5 min post $H_2O_2$ treatment). Cells were kept in the dark for 90 min (green) and were then transferred back to the light (cyan). The same $H_2O_2$ treatment without transition to the dark (light ctrl, black) and maximum oxidation (200 μM $H_2O_2$, red) and reduction (2 mM DTT, blue) are shown for reference. The experiment was done in triplicates that were highly similar, for visualization the first replica is shown. Each histogram consists of >8000 cells. (D) Mean ± SEM basal (control) chl-roGFP OxD over time of cells transitioned to the dark for 90 min (gray box) at time 0 ('dark') and cells kept in light conditions ('light'), n = 3 biological repeats. Pre-treatment baseline under light conditions is shown. SEM lower than 0.5% are not shown. (E) The fraction of dead cells 24 hr post $H_2O_2$ treatment, with or without transition to the dark ('dark' and 'light' respectively), as measured by positive Sytox staining. Data is shown as mean ± SEM, n = 3 biological repeats. P values: *=0.0064, **=0.0026, ***=$2 \cdot 10^{-5}$, t-test.

DOI: https://doi.org/10.7554/eLife.47732.031

The following source data and figure supplements are available for figure 7:

**Source data 1.** Flow cytometry measurements of chl-roGFP oxidation in response to dark and $H_2O_2$ treatments.
DOI: https://doi.org/10.7554/eLife.47732.034

**Source data 2.** Cell death in response to dark and $H_2O_2$ treatments.
DOI: https://doi.org/10.7554/eLife.47732.035

**Figure supplement 1.** Redox response of chl-roGFP to transition to the dark.

*Figure 7 continued on next page*

*Figure 7 continued*

DOI: https://doi.org/10.7554/eLife.47732.032

**Figure supplement 2.** Responses of chloroplast and nucleus targeted roGFP to transition to the dark.

DOI: https://doi.org/10.7554/eLife.47732.033

Analysis of the redox-sensitive proteome in *P. tricornutum* revealed over-representation of chloroplast-targeted proteins, that were also oxidized to a greater degree under $H_2O_2$ treatment as compared to other subcellular compartments, further supporting the existence of a redox-based signaling network in the chloroplast (*Rosenwasser et al., 2014*; *Woehle et al., 2017*).

The role of chloroplast $E_{GSH}$ perturbations in sensing specific stress cues gains further support by the early chl-roGFP oxidation, which preceded the 'point of no return', after which cell death was irreversibly activated in the 'oxidized' subpopulation (*Figure 3B*). This 'pre-commitment' phase provides an opportunity for cells to recover if conditions change during a narrow time frame of ~30–60 min following oxidative stress in most cells (*Figure 3B*), before the cell has accumulated damage beyond repair or a PCD cascade was fully activated. This 'pre-commitment' phase was shown previously in diatoms, as exogenous application of the antioxidant GSH rescued cells from otherwise lethal treatments of infochemicals or $H_2O_2$ only during the first hour (*Graff van Creveld et al., 2015*; *Volpert et al., 2018*). These findings shed light on the timeline of events in PCD progression in diatoms.

To date, the role of the chloroplast in mediating PCD has remained elusive, although mitochondria-generated ROS are known to play a key role in PCD in plants and animals (*Van Aken and Van Breusegem, 2015*; *Lam et al., 2001*). This knowledge gap is even greater in unicellular marine algae, for which the molecular basis for the PCD machinery is largely unknown (*Bidle, 2016*). In *P. tricornutum*, early mitochondrial oxidation was shown to precede subsequent cell-death in response to various stress conditions at the population level, but the link with chloroplast $E_{GSH}$ was less clear and depended on the specific stress cue (*Graff van Creveld et al., 2015*). In another diatom, chloroplast $E_{GSH}$ was shown to mediate changes in oxidative stress sensitivity upon light-dark transitions (*Volpert et al., 2018*). A recent model in plants suggested possible mitochondria-chloroplast cooperative interactions in the execution of ROS-mediated PCD (*Van Aken and Van Breusegem, 2015*). In addition, there are evidence for energetic coupling of chloroplasts and mitochondria in diatoms, and they use extensive energetic exchanges between these organelles to regulate the ATP/NADPH ratio (*Bailleul et al., 2015*). Taken together with the results presented here, we suggest that redox dynamics of both the mitochondria and the chloroplast are involved in cell fate regulation in diatoms.

We propose that cells that accumulate ROS above a certain threshold are likely to induce cell death with PCD-like hallmarks (*Figure 9*), as was shown in response to $H_2O_2$ (*Graff van Creveld et al., 2015*) and as observed in the death of the 'oxidized' subpopulation (*Figures 3B* and *4F-G*, and *Figure 8D*). Cells that do not cross this threshold are able to recover and acclimate, as in the 'reduced' subpopulation (*Figures 3B* and *4F-G*, and *Figure 8D*). Based on the data from the microfluidics setup, which allowed cell tracking throughout the entire dynamics during exposure to $H_2O_2$, we propose that such a 'death threshold' could be detected by early chl-roGFP oxidation (*Figure 4G*). This 'death threshold' may be dependent on additional factors, such as ecological context or the specific stressor. The balance between the cellular metabolic state, antioxidant capacity, and the magnitude of the applied stress determines whether a cell will cross the 'death threshold', leading to a differential response within the population. Harsher stress conditions will have a stronger effect on the population, leading to more cells crossing the threshold and exhibiting early oxidation and subsequent cell death, as shown with increasing $H_2O_2$ doses (*Figure 2M*) or prolonged exposure to HL (*Figure 8B*).

The source for the cell-to-cell variability observed in our system is yet to be further explored, but the results provide insights into factors that may drive it. Since clonal populations originating from single-cell isolates maintained the bi-stable chloroplast response, the variability does not result from genetic differences but rather from phenotypic plasticity (*Figure 5*). The combination of factors such as life history (*Graff van Creveld et al., 2016*; *Murik et al., 2014*), cell cycle phase (*Papagiannakis et al., 2017*; *Mathis and Ackermann, 2016*), cell age (*Levy et al., 2012*; *Radzinski et al., 2018*), metabolic activity (*Şimşek and Kim, 2018*; *Campbell et al., 2018*),

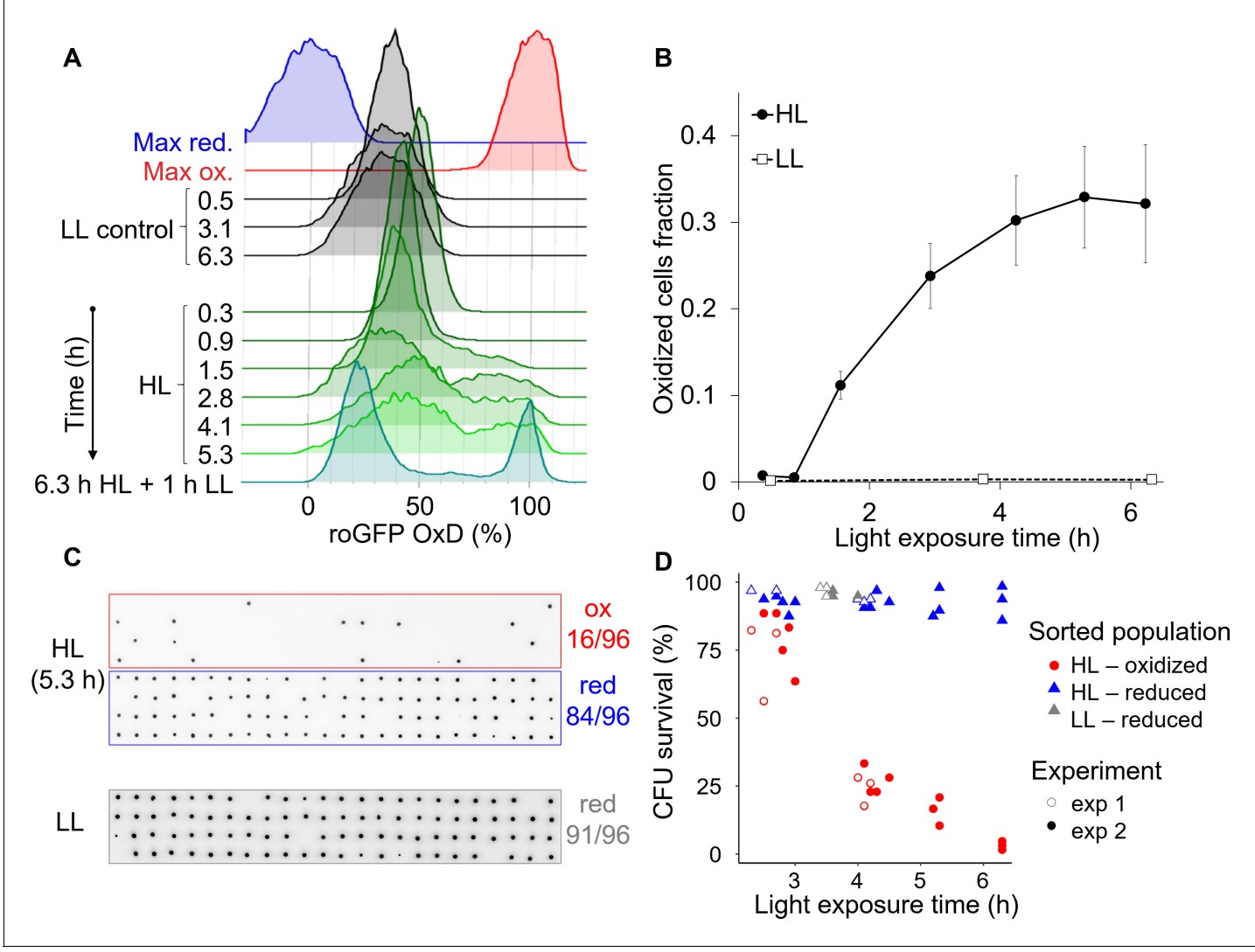

**Figure 8.** High light induces the emergence of an 'oxidized' subpopulation with decreased survival. Cells expressing chl-roGFP were exposed to either high light (HL, 2000 μmol photons m$^{-2}$ s$^{-1}$) or low light (LL, 14 μmol photons m$^{-2}$ s$^{-1}$), and were FACS sorted based on their chl-roGFP oxidation at different times of light exposure to measure single-cell survival using the colony forming units (CFU) assay. (**A**) Flow cytometry measurements of chl-roGFP OxD distribution in the population over time of HL exposure. Cells were kept in HL for 6.3 hr (green) and were then transferred back to LL for 1 hr (cyan). LL control (black) and maximum oxidation (200 μM H$_2$O$_2$, red) and reduction (2 mM DTT, blue) are shown for reference. The experiment was done in triplicates, for visualization the first replica is shown. Each histogram consists of >9000 cells. (**B**) The fraction of oxidized chl-roGFP cells over time of exposure to HL. Mean ± SEM, N = 3. (**C**) Images of chlorophyll auto-fluorescence of colonies generated from single cells that were sorted based on chl-roGFP oxidation following exposure to HL or LL for the CFU assay. Panels from top to bottom: oxidized (red box) and reduced (blue box) subpopulations sorted after 5.3 hr of HL, reduced subpopulation sorted under LL control conditions (gray box). CFU survival (%) was measured by the number of colonies divided by number of sorted cells, as shown at the right of each panel. Gates used for sorting are shown in *Figure 8—figure supplement 1*. (**D**) CFU survival (%) of single cells that were sorted based on chl-roGFP oxidation at different times of light exposure onto agar plates. Data of 2 independent experiments each with three biological repeats, 64–96 sorted cells per repeat, individual repeats are shown. (**A–D**) Data points of 6.3 hr HL exposure were exposed to HL for 6.3 hr and then moved back to LL for 1 hr.

DOI: https://doi.org/10.7554/eLife.47732.036

The following source data and figure supplements are available for figure 8:

**Source data 1.** The fraction of oxidized cells over time in response to high light.
DOI: https://doi.org/10.7554/eLife.47732.039
**Source data 2.** Colony forming units survival of single cells sorted following high light exposure based on chl-roGFP oxidation.
DOI: https://doi.org/10.7554/eLife.47732.040
**Figure supplement 1.** High light induces the emergence of an oxidized subpopulation.
DOI: https://doi.org/10.7554/eLife.47732.037

*Figure 8 continued on next page*

*Figure 8 continued*

**Figure supplement 2.** Measurements of nuclear roGFP oxidation in response to high light.

DOI: https://doi.org/10.7554/eLife.47732.038

heterogeneous microenvironment (*Stocker, 2012*) and biological noise (*Balaban, 2011*; *Raj and van Oudenaarden, 2008*) results in a distribution of different metabolic states within the population (*Ackermann, 2015*; *Takhaveev and Heinemann, 2018*), which could lead to differential sensitivity to oxidative stress. In yeast for example, redox-based heterogeneity was linked to proliferation and aging (*Radzinski et al., 2018*). It remains to be investigated whether the emergence of the subpopulations represents heterogeneity that occurs following exposure to stress, or rather a pre-existing variability within the population. Nevertheless, differences in cell-cycle phase distribution between the subpopulations likely represent pre-existing disparities, supporting the latter. However, differences in cell cycle do not completely explain the variability, and are likely to be a contributing factor rather than the source, for example by antioxidants oscillations (*Papagiannakis et al., 2017*; *Burhans and Heintz, 2009*; *Mathis and Ackermann, 2016*; *Diaz-Vivancos et al., 2015*). In addition, the microfluidics experiments showed no clear differences in chl-roGFP OxD between the subpopulations prior to the treatment (*Figure 4—figure supplement 2B*), demonstrating that the possible pre-existing heterogeneity is not reflected in the chloroplast $E_{GSH}$ basal level, but rather is based on a different parameter that is yet to be identified.

Unlike in previous studies conducted on heterogeneity, the mechanism that generates variability in our system is light-dependent, as the bi-stable chl-roGFP pattern was abolished when the cells were under darkness and the entire population became more sensitive to oxidative stress (*Figure 7A–C*). The antioxidant capacity of a diatom cell depends on photosynthesis-generated NADPH, which is also used for GSH recycling. The transition to the dark may have compromised the biosynthesis and recycling of GSH, therefore enhancing sensitivity to oxidative stress, as was shown in another diatom (*Volpert et al., 2018*). Taken together, we propose that the source for

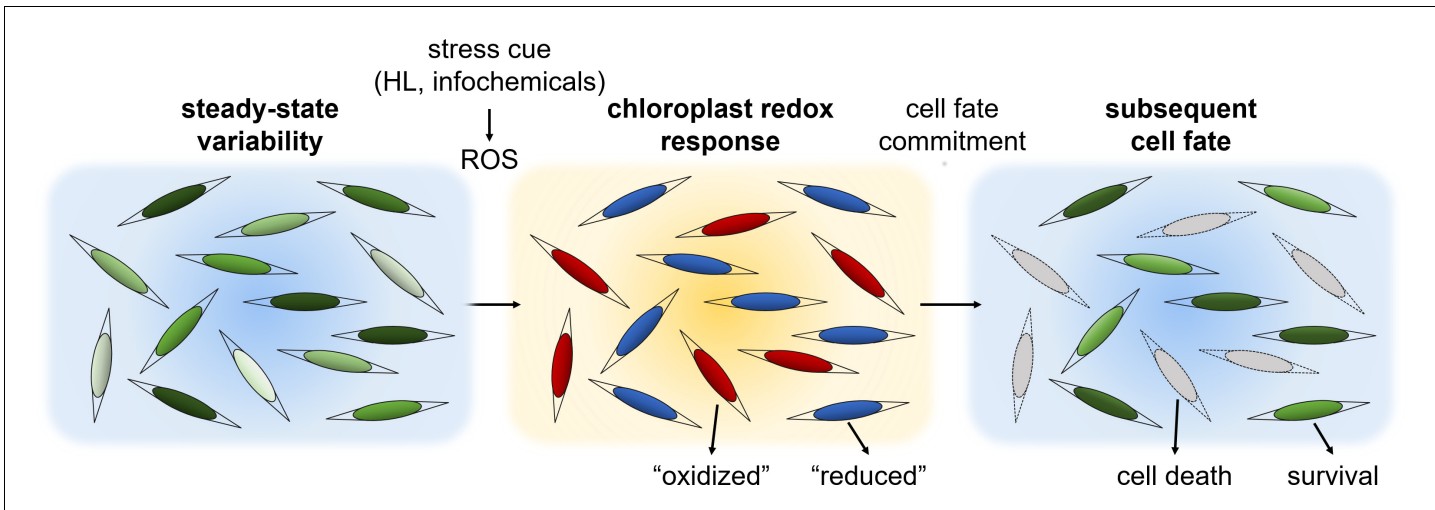

**Figure 9.** A conceptual model: phenotypic variability within diatom populations affects cell fate determination in response to stress conditions. At steady state conditions, cells within the population have diverse metabolic states due to various factors such as local ROS levels, antioxidant capacity, metabolic activity, growth phase and cell cycle position. We propose that this variability could lead to a differential response to environmental stressors. Exposure to specific stress conditions, such as high light (HL) or toxic infochemicals, can lead to ROS accumulation at different subcellular compartments, including the chloroplast, which is used to sense the stress cue and regulate cell fate. Cells at a more susceptible metabolic state may accumulate high ROS levels and will subsequently die, as observed in the 'oxidized' subpopulation. More resilient cells may exhibit milder oxidation and may be able to acclimate, as observed in the 'reduced' subpopulation. Chloroplast $E_{GSH}$ oxidation is an early stage in this stress response and precedes the commitment to cell fate.

DOI: https://doi.org/10.7554/eLife.47732.041

heterogeneity could be variability in the flux of photosynthesis-derived reductive power, regulating the recycling rates of antioxidants.

The light-dependent emergence of distinct subpopulations and the increased sensitivity under dark suggest important implications for environmental scenarios. Fluctuating light conditions are frequent in natural environments due to mixing, shading, the diel cycle, and tide (in coastal and intertidal regions), and may greatly affect diatoms' susceptibility to diverse abiotic stresses and biotic interactions with pathogens. Phenotypic variability can provide an important strategy to cope with fluctuating environments in microbial populations (*Ackermann, 2015*). Co-existence of subpopulations with different susceptibilities to specific stressors can be viewed as a 'bet-hedging' strategy of the population, enabling at least a portion of the population to survive unpredicted stress events and subsequently leads to a growth benefit at the population level (*Schreiber et al., 2016*; *Sengupta et al., 2017*; *Levy et al., 2012*; *Ackermann, 2015*). In diatoms, phenotypic variability in cell size, shape and susceptibility to stress conditions was suggested (*Armbrust, 2009*; *Graff van Creveld et al., 2015*; *Volpert et al., 2018*; *De Martino et al., 2011*), but until now the experimental setups were not designed to study individuality in stress response. Future studies are required to investigate the possible tradeoff involved in maintaining high antioxidant capacity. For example, resilience to oxidative stress may come with a cost in the ability to sense environmental cues with high precision, as high ROS buffering capacity may mask milder ROS cues (*Woehle et al., 2017*). Redox-based phenotypic variability may provide a rapid and adjustable strategy to cope with unpredicted stress conditions as compared to relying only on genetic diversity.

The novel approaches developed here provide new insights into individuality in marine phytoplankton, and enable studying dynamic processes at the single-cell level in diatoms and possibly other ecologically relevant microorganisms. The mechanisms that underlie differential sensitivity to oxidative stress are yet to be explored. The findings presented here show promising ecological implications for light-dependent heterogeneity, and future studies will unravel its ecological significance in the marine environment.

## Materials and methods

**Key resources table**

| Reagent type (species) or resource | Designation | Source or reference | Identifiers | Additional information |
|---|---|---|---|---|
| Strain, strain background (*Phaeodactylum tricornutum*) | Pt1 8.6 (WT) | National Center of Marine Algae and Microbiota (NCMA) | CCMP2561 | |
| Strain, strain background (*Phaeodactylum tricornutum*) | chl-roGFP | (*Rosenwasser et al., 2014*) | | |
| Strain, strain background (*Phaeodactylum tricornutum*) | nuc-roGFP | (*Rosenwasser et al., 2014*) | | |
| Strain, strain background (*Phaeodactylum tricornutum*) | mit-roGFP | (*Rosenwasser et al., 2014*) | | |
| Commercial assay or kit | Sytox Green | Invitrogen | | |
| Software, algorithm | Image analysis MATLAB script | This paper and on GitHub (*Mizrachi, 2018*): https://github.com/aviamiz/ITRIA | | |

### Culture growth

*P. tricornutum* accession Pt1 8.6 (CCMP2561 in the Provasoli-Guillard National Center for Culture of Marine Phytoplankton) was purchased from the National Center of Marine Algae and Microbiota

(NCMA, formerly known as CCMP). Cultures were grown in filtered seawater (FSW) supplemented with F/2 media (*Guillard and Ryther, 1962*) at 18°C with 16:8 hr light:dark cycle and 80 µmol photons m$^{-2}$ sec$^{-1}$ light intensity supplied by cool-white LED lights (Edison, New Taipei, Taiwan). Strains expressing roGFP were obtained as described previously (*Graff van Creveld et al., 2015*; *Rosenwasser et al., 2014*). Cultures were kept in exponential phase (<2·10$^6$ cells·ml$^{-1}$) for at least 1 week and were sequentially diluted at least three times prior to experiments, experiments were performed in ~0.5–1·10$^6$ cells·ml$^{-1}$. All cultures were counted and diluted a day before the experiment to ensure the same cell concentrations between samples. Cell concentration was measured using Multisizer 4 COULTER COUNTER (Beckman Coulter).

## roGFP measurements

roGFP oxidation was measured over time following the addition of H$_2$O$_2$ or in untreated control using the ratio between two fluorescence channels, i405 and i488, by fluorescence microscopy (described below) and by flow cytometry using BD LSRII analyzer, BD FACSAria II and BD FACSAria III. The roGFP ratio (i405/i488) increases upon oxidation of the probe (*Schwarzländer et al., 2008*). The oxidation degree of roGFP (OxD) was calculated according to Schwarzländer et al. (*Schwarzländer et al., 2008*):

$$OxD_{roGFP} = \frac{R - R_{red}}{\frac{i488_{ox}}{i488_{red}}(R_{ox} - R) + (R - R_{red})}$$

Where R is the roGFP ratio of i405/i488, R$_{red}$ is the ratio of fully reduced form (15–50 min post treatment with 2 mM Dithiothreitol, DTT), R$_{ox}$ is the ratio of the fully oxidized form (7–30 min post treatment with 200 µM H$_2$O$_2$), and i488$_{ox}$ and i488$_{red}$ are the i488 of the maximum oxidized and maximum reduced forms respectively. For sorting purposes, roGFP ratio was used, as exact OxD cannot be calculated prior to sorting, both parameters give a similar partition between the subpopulations (data not shown). In flow cytometry measurements, i405 was measured using excitation (ex) 407 nm, emission (em) 530/30 nm or 525/25 nm, and i488 was measured using ex 488 nm, em 530/30 nm. Only roGFP positive cells (roGFP+) which had a clear roGFP fluorescence signal separated from WT auto-fluorescence (AF) were included in the analysis. roGFP+ gate was determined either by roGFP relative expression level, which was measured by multiplication of i405 and i488 (*Figure 2—figure supplement 2D*), or based on i405 and i488 intensity in sorting experiments, as relative roGFP expression could be calculated only post acquisition (see *Figure 8—figure supplement 1* for example). Dynamic range of roGFP was calculated by the ratio of R$_{ox}$/R$_{red}$ (*Table 1*). For H$_2$O$_2$ treatments, H$_2$O$_2$ was added at time 0 from a freshly prepared 20 mM stock to *P. tricornutum* cultures to a final concentration of 5–200 µM. Flow cytometry measurements were done under ambient light (5.5–14 µmol photons m$^{-2}$ sec$^{-1}$) and temperature (20–22°C) conditions, unless stated otherwise. For dark treatment samples were covered with aluminum foil. High light of 1,700–2,200 µmol photons m$^{-2}$ sec$^{-1}$ was applied using a LED lamp (deviations are due to uneven illumination depending on the distance from the lamp center). To avoid heating by the HL lamp, samples were kept at 16°C using a chilled stage (LCI, Korea).

**Table 1.** Dynamic range of roGFP targeted to different organelles.
The dynamic range (R$_{ox}$/R$_{red}$) was calculated by dividing the mean roGFP ratio under maximum oxidation conditions (200 µM H$_2$O$_2$) by that under maximum reduction conditions (2 mM DTT) using data from flow cytometry and microfluidics imaging experiments.

| *P. tricornutum* strain | Flow cytometry dynamic range | Microfluidics imaging dynamic range |
|---|---|---|
| chl-roGFP | 5.57 | 5.42 |
| nuc-roGFP | 5.94 | - |
| mit-roGFP | 3.50 | - |

DOI: https://doi.org/10.7554/eLife.47732.042

## Cell death analysis

For cell death analysis, samples were stained with a final concentration of 1 µM Sytox Green (Invitrogen), incubated in the dark for 30–60 min at RT and analyzed using an Eclipse flow cytometer (ex 488 nm, em 525/50 nm). Unstained samples were used as control to discriminate background signal. In microfluidics experiments, Sytox was dissolved in fresh media (FSW + F/2) to a final concentration of 1–2 µM, and inserted into the system at 21.5–23 hr post treatment, without changing the flow rate (1 µl/min). Fresh stain was continuously flowing through the system for at least 1.5 hr during which cells were imaged for Sytox staining (ex 470/40 nm, em 525/50 nm) as described below. For Sytox staining analysis, images of 30–106 min incubation time were used based on highest staining and best focus.

## Sorting of subpopulations and generation of clonal populations

To measure survival and generate clonal populations originating from different subpopulations, cells expressing chl-roGFP of the 'oxidized' and 'reduced' subpopulations were sorted according to their roGFP ratio at different times post 80 µM $H_2O_2$ treatment or HL using BD FACS AriaII and BD FACS AriaIII. The 'oxidized' and 'reduced' subpopulations gates were based on visible separation and avoiding cells near intermediate values. Untreated roGFP+ control cells were sorted based on clear separation of roGFP fluorescence from WT AF as described above (the gate upstream of the subpopulations' gates), regardless of roGFP oxidation. To avoid oxidation due to darkness within the FACS machine, sorting times were minimized and the 'oxidized' subpopulation was sorted first followed immediately by sorting of the 'reduced' subpopulation. However, in longer sorting sessions as for the Sytox and cell cycle analyses, some cells within the sorted 'oxidized' subpopulation may have been oxidized due to the combined effect of $H_2O_2$ and exposure to the dark within the FACS. For Sytox analysis of cell death post sorting, 10,000 cells/well were sorted into fresh media (FSW + F/2) in triplicate biological repeats. For single cell survival 1 cell/well was sorted into 96-well plates containing either 'agar' (1.5% agarose +FSW/2 + F/2 + antibiotics) or 'liquid' (FSW + F/20) fresh media. For the HL CFU assay, 384 cells were sorted onto one-well agar plates. Cells grown in liquid were further diluted and then spotted on agar plates. 3.5–9 weeks post sorting colonies were counted manually ($H_2O_2$ experiments) or scanned using Amersham Typhoon 5 Biomolecular Imager and quantified using ImageQuant (HL experiments), assuming each colony originates from a single surviving cell. CFU survival was calculated as CFU number divided by the number of sorted cells. Since survival was highly similar in liquid and in agar the results of these two methods were combined together. Each method was done in biological triplicates per medium type per experiment. In *Figure 3B* data is shown for two independent experiments for time points 30 and 100 min and one experiment for the 60 min time point. For generation of clonal populations, single cells sorted into liquid medium were used. Clones were exposed to 80 µM and 100 µM $H_2O_2$ ~ 3–6 weeks post sorting, and their chl-roGFP oxidation was measured using flow cytometry. A total of 18 'control', 29 'oxidized' and 32 'reduced' clones were examined in two independent experiments.

## Microfluidics chip preparation

Microfluidics chip design was based on Shapiro et al. (*Shapiro et al., 2016*) and was modified for *P. tricornutum* cells. Each chip contained 4 channels of 2 cm length X 0.2 cm width X 150 µm height with one circular widening of 0.4 cm diameter, with a total volume of ~12.7 µl per channel (*Figure 4—figure supplement 1B–C*). The microfluidics chip was etched into a silicone elastomer (SYLGARD 184, Dow Corning) using soft lithography. Silicone elastomers were prepared by mixing the two components in a 10:1 ratio and were poured onto the dust-free wafer, de-aired in a desiccator to eliminate air bubbles, and incubated overnight at 60°C for curing to generate the Polydimethylsiloxane (PDMS) microfluidics chips. Inlet and outlet holes were punched at both ends of each channel using a 1 mm biopsy punch (AcuDerm, FL, USA). The PDMS chip was placed on the clean surface of a new glass microscope 60 × 24 mm cover slip using plasma bonding with a BD-20AC corona treater (Electro-Technic Products) followed by heating of 100°C for >15 min to ensure covalent bonding of the PDMS and the glass.

## Microfluidics live imaging

Light and epifluorescence microscopy imaging was performed using a fully motorized Olympus IX81 microscope (Olympus) equipped with ZDC component for focus drift compensation, 20X air objective (numerical aperture 0.5), and Lumen 200PRO illumination system (Prior Scientific). Images were captured using a Coolsnap HQ2 CCD camera (Photometrics, Tuscon, AZ, USA). The microfluidics chip was mounted on a motorized XY stage (Prior Scientific, MA, USA) with a temperature-controlled inset (LCI, Korea) set to 18˚C (*Figure 4—figure supplement 1D*). The outlet tubes were connected to syringe pumps (New Era Pump Systems, NY, USA) set to withdrawal mode, using negative pressure for flow generation. The inlet tubes were connected to Eppendorf reservoirs, containing the fluid to be inserted into the system. Experiment layout is shown in *Figure 4—figure supplement 1A*. Chambers were washed with at least 500 µl of pure ethanol, then double-distilled water and then fresh media prior to the introduction of cells. Cells were introduced into the system and settled on the glass bottom.

Flow rate was kept at 1 µl/min for the duration of the experiment, except during cell introduction (100 µl/min), cell settlement (up to 20 µl/min with occasional stops), and treatment introduction (10 µl/min for the initial 10 min for rapid replacement of media). Following settlement and at least 1 hr after cells were introduced to the system, cells were imaged for roGFP measurements (roGFP i405: ex 405/20 nm, em 525/50 nm; roGFP i488: ex 470/40 nm, em 525/50 nm), chlorophyll auto-fluorescence (ex 470/40 nm; em 590 lp) and bright field (BF, without a condenser). Each chip contained four chambers that were imaged sequentially: chl-roGFP control, chl-roGFP 80 µM $H_2O_2$ treated, WT 80 µM $H_2O_2$ treated and WT control (WT strain can be used to monitor auto-fluorescence changes and leakage during experiments). In each chamber, 5–6 different fields were imaged every 20 min over the course of >24 hr to avoid photo-toxicity. Ambient light was provided during light period using the microscope's BF illumination without a condenser, light intensity ranging between 34 (at the very edge, outside the imaging region) to 80 (center) µmol photons $m^{-2}$ $sec^{-1}$. No images were obtained during the night to avoid disturbance to the diurnal cycle. After imaging the basal state of the cells, treatments of either 80 µM $H_2O_2$ dissolved in fresh media (FSW + F/2) or fresh media control were introduced to the system continuously for ~2.5–3 hr, after which they were gradually washed away by fresh media. To quantify cell death, Sytox green was introduced into the system at 21.5–23 hr post treatment (see above) and was imaged using the roGFP i488 channel with a shorter exposure time. The Sytox signal was separated from the roGFP i488 based on its stronger fluorescence and localization to the nucleus. Only a small fraction of cells within the control treatment were Sytox positive (0.0054%), and in addition some control and 'reduced' cells proliferated during the experiments (*Videos 1–2*), indicating that cells remained viable in this experimental setup.

## Image analysis

Image analysis was performed using a designated MATLAB based script (see overview in *Figure 4—figure supplement 4*) that is available on GitHub: https://github.com/aviamiz/ITRIA (*Mizrachi, 2018*; copy archived at https://github.com/elifesciences-publications/ITRIA). Images were imported using bio-formats (*Linkert et al., 2010*). Then, image registration for XY drift correction was done using the Image Stabilizer plugin (*Li, 2008*) for FIJI (Fiji Is Just ImageJ) and using MIJI (*Prodanov and Tinevez, 2012*) to access FIJI from MATLAB. Then images were normalized by bit-depth. Background subtraction was done based on mean value of a user-defined region of interest (ROI) that did not include cells. All fluorescence channels (i405, i488 and chlorophyll) were thresholded by a user-defined value to generate masks of positive expression. The roGFP relative expression level was calculated pixel-by-pixel by multiplication of the i405 and i488, only at pixels that were co-localized in the i405 and i488 masks. Then, roGFP relative expression (i405 * i488) was thresholded in order to include only pixels with high enough signal, based on a user-defined threshold. The roGFP ratio and OxD were calculated pixel-by-pixel as described above, pixels that were not included in the roGFP expression mask were excluded and set to NaN (not a number). For values of maximum oxidation and reduction of roGFP, cells were imaged in the same microfluidics imaging setup following treatments of 200 µM $H_2O_2$ and 2 mM DTT respectively (see 'roGFP calculations'). Cell segmentation was based on i405 (chl-roGFP strain) or chlorophyll (WT strain) masks and fluorescence intensity using watershed transformation. Cells were filtered based on area, major and minor

axis length, and eccentricity in order to exclude clumps of cells and doublets. Cell tracking was adapted and modified from a MATLAB code kindly provided by Vicente I. Fernandez and Roman Stocker (*Smriga et al., 2016*; *Shapiro et al., 2014*). In short, particles were tracked based on minimizing the distance between particle centroids in adjacent frames within a distance limit. Sytox analysis was based on a user defined threshold and co-localization of the Sytox with the extended cell region within the cell segmentation mask. Images from the same experiment were analyzed using the same values for all thresholds and parameters, except for Sytox analysis in which the threshold was adjusted manually to validate correct assignment of cell-fate and to avoid effects of focus differences. Cells that were not detected in the frame used for Sytox analysis or were not tracked for at least six consecutive frames were excluded from further analysis. The 74% OxD threshold used for early discrimination between cells that subsequently died or survived (*Figure 4E–G* and *Figure 4—figure supplement 2*) was based on logistic regression modeling of cell fate at the end of the experiments as a function of chl-roGFP OxD 40 min post 80 μM $H_2O_2$ treatment (*Figure 4—figure supplement 3*). The observed roGFP OxD of more than 100% oxidation in some cells could result from increased auto-fluorescence leakage to the i405 channel at later times post treatment (see *Figure 2—figure supplements 5–6*).

## Cell cycle analysis

Cell cycle analysis was based on *Huysman et al. (2010)* and modified for sorted cells. 30,000 cells of 'oxidized' and 'reduced' sub-populations were sorted 30 min post 80 μM $H_2O_2$ treatment into 260 μl 80% ethanol kept at 5 ˚C, reaching a final concentration of 70% ethanol. Control untreated cells and synchronized cells for reference (20 hr dark, as previously described; *Huysman et al., 2010*) were sorted based on positive roGFP fluorescence. Cells were then gently mixed and kept at 4 ˚C until further processing. Then, 500 μl of 0.1% bovine serum albumin in phosphate-buffered saline (PBS) was added to improve pellet yield, and samples were centrifuged at 4000 rcf at 4 ˚C for 10 min to discard supernatant. Cells were then washed with PBS, re-suspended and stained with 4',6-diamidino-2-phenylindole (DAPI, Sigma) at a final concentration of 10 ng/ml. Samples were analyzed using BD LSRII analyzer, with ex 355 nm and em 450/50 nm. Synchronized cells were used as a reference to validate the gates for $G_1$ and $G_2$/M phases (data not shown). S phase was not clearly detected in this analysis.

## Statistics

All statistical analyses were done in R. ANOVA or ANCOVA were used for multiple comparisons, and then Dunnett test or Tukey test were performed were applicable. For comparisons of two samples, t-test was used, and paired t-test was used where applicable. Values are represented as mean ± SEM unless specified otherwise. Box-plot was generated using the web tool BoxPlotR http://shiny.chemgrid.org/boxplotr/ (*Spitzer et al., 2014*) using Tukey whiskers, which extend to data points that are less than 1.5 x Interquartile range away from 1 st/3rd quartile.

## Data availability

All relevant data supporting the findings of the study are available in this article and its Supplementary Information, or from the corresponding author upon request.All data generated or analysed during this study are included in the manuscript and supporting files. Source data files have been provided for Figures 1– 8. MATLAB script used for image analysis is available at GitHub, as referenced in the methods section: https://github.com/aviamiz/ITRIA.

## Acknowledgements

We thank Dr. Ron Rotkopf from the Bioinformatics Unit at the Life Sciences Core Facilities, Weizmann Institute of Science for assisting with the statistical analysis. We thank Dr. Vicente I Fernandez and Prof. Roman Stocker for help with the cell tracking algorithm. We thank Dr. Uri Sheyn and the flow cytometry unit at the Life Sciences Core Facilities, Weizmann Institute of Science for technical help with FACS operation. We thank Jenny Mizrahi for proofing the manuscript. This research was supported by the Israeli Science Foundation (ISF) (grant #712233) awarded to AV.

## Additional information

### Funding

| Funder | Grant reference number | Author |
| --- | --- | --- |
| Israel Science Foundation | 712233 | Assaf Vardi |

The funders had no role in study design, data collection and interpretation, or the decision to submit the work for publication.

### Author contributions

Avia Mizrachi, Conceptualization, Data curation, Formal analysis, Validation, Investigation, Visualization, Methodology, Writing—original draft, Code development for image analysis, Project administration; Shiri Graff van Creveld, Conceptualization, Data curation, Formal analysis, Visualization, Methodology, Writing—review and editing; Orr H Shapiro, Visualization, Methodology, Writing—review and editing; Shilo Rosenwasser, Conceptualization, Formal analysis, Visualization, Methodology, Writing—review and editing; Assaf Vardi, Conceptualization, Resources, Formal analysis, Supervision, Funding acquisition, Investigation, Visualization, Writing—original draft, Project administration

### Author ORCIDs

Avia Mizrachi (iD) https://orcid.org/0000-0001-7724-9275
Shiri Graff van Creveld (iD) https://orcid.org/0000-0002-3445-3046
Orr H Shapiro (iD) https://orcid.org/0000-0002-3222-9809
Shilo Rosenwasser (iD) https://orcid.org/0000-0001-8565-9979
Assaf Vardi (iD) https://orcid.org/0000-0002-7079-0234

### Decision letter and Author response

Decision letter https://doi.org/10.7554/eLife.47732.045
Author response https://doi.org/10.7554/eLife.47732.046

## Additional files

### Supplementary files

• Transparent reporting form
DOI: https://doi.org/10.7554/eLife.47732.043

### Data availability

All data generated or analysed during this study are included in the manuscript and supporting files. Source data files have been provided for Figures 1– 8. MATLAB script used for image analysis is available at GitHub, as referenced in the methods section: https://github.com/aviamiz/ITRIA. (copy archived at https://github.com/elifesciences-publications/ITRIA).

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
