## [Decision Letter]

[Editors’ note: a previous version of this study was rejected after peer review, but the authors submitted for reconsideration. The first decision letter after peer review is shown below.]

Thank you for submitting your work entitled "Single-cell heterogeneity in the chloroplast redox state mediates acclimation to stress in a marine diatom" for consideration by *eLife*. Your article has been reviewed by three peer reviewers, and the evaluation has been overseen by a Reviewing Editor and a Senior Editor. The following individual involved in review of your submission has agreed to reveal their identity: E. Virginia Armbrust (Reviewer #1).

Our decision has been reached after consultation between the reviewers. Based on these discussions and the individual reviews below, we regret to inform you that your work will not be considered further for publication in *eLife* at this time.

While there is much to like about the technical approach used in this study and the rigor of its execution, whether the conditions used in these studies are representative of natural conditions, and thereby meaningful outside of the laboratory, is unclear. This mitigates enthusiasm for publishing the story in *eLife* in its current form. Which environmental stressors generate H_2_O_2_ intracellularly at the levels used in these studies? How do these stressors stimulate intracellular H_2_O_2_ production? The topic of how marine diatoms respond to environmental stress is interesting, but the authors have more work to do to establish that the conditions used in this work are environmentally and physiologically representative. If this can be done, we encourage a revised version of the manuscript to be resubmitted to *eLife*. We recognize that providing satisfying answers to these questions is non-trivial, this we are rejecting this version of the story to provide the authors sufficient time to address them. If the authors elect to perform more work and resubmit, we will send the manuscript back out to the same reviewers in the future.

*Reviewer #1:*

The manuscript by Mizrachi et al. is a carefully crafted documentation of phenotypic heterogeneity within the model diatom *Phaeodactylum tricornutum*. Their target phenotype is responsiveness to exogenously added H_2_O_2_, which serves as a surrogate for oxidative stress. The authors rely on their previously developed roGFP sensors for plastid, mitochondrial, and nuclear oxidation state to show that two phenotypic variants in plastid oxidation are found within exposed populations of cells. Those that display greater oxidative stress are strongly correlated with subsequent cell death. This manuscript combines multiple state of the art methods to study individual cell phenotypes including the use of the roGFP tags, flow cytometry sorting, and microfluidics. Phenotypic variability between individual cells is emerging as an important feature that influences how populations of cells respond to environmental change. I have relatively few suggestions for further improvement of the manuscript.

1) Overall, the manuscript is well-written, although it was a little repetitive in sections. This is a manuscript packed with data and figures. Removing any repetitiveness would help with the flow. One example is where the authors state:

"To further explore the physiological factors that may affect the bi-stable distribution, we investigated the effects of different growth phases on oxidation of chl-roGFP in response to H_2_O_2_. Since aging has been associated with changes in redox states in several model systems^40-42^, we hypothesized that culture growth phase may affect the susceptibility of the population to oxidative stress."

This could be simplified to:

"To further explore the physiological factors that may affect the bi-stable distribution, we investigated the effects of different growth phases on oxidation of chl-roGFP in response to H_2_O_2_ as aging has been associated with changes in redox states in several model systems^40-42^."

2) Introduction, second paragraph – typo: "response" should be replaced with "respond".

3) Materials and methods – growth state is clearly an important parameter for phenotypic variability. The authors should provide a bit more information than "Cultures were kept in exponential phase." How were cultures maintained in exponential phase? This is particularly important for interpreting Figure 6A as the initial slopes appears to differ between the different curves.

4) Materials and methods – There are a number of places within the Materials and methods that include results (see for example, in the subsection “roGFP measurements”) and references to figures that are described in the main text. This can be confusing for the reader and I suggest that the authors remove references to figures and results from the Materials and methods section.

5) Figure 1A, control. Could the authors comment on the slight shift to a more reduced state at 50 minutes and then the return to the initial "control" state at 282 minutes? Could this reflect cell cycle transitions?

6) Figure 1. The authors could consider adding a control distribution to panels 1B-D in a manner similar to what was done with the distributions for fully oxidized and reduced. This would help illustrate any shifts away from/return to the control state over the course of the experiment.

7) Figure 2B. The schematic panel is not required as part of main text.

8) Figure 2C legend. I realize this was discussed in the Materials and methods, but in the legend, it is not clear what is meant by "sorted based on positive roGFP fluorescence" given that the sort window is indicated in the 2C panel.

9) The use of colony forming units to detect survival was initially confusing to me. I realized after reading further that this was done to allow the authors to generate clonal cell lines for subsequent analysis. This could be clarified when first referenced in the main text.

10) Figure 3H (the diagram) should be moved to the supplement rather than included as part of the main figure. In fact, this panel is not currently referenced in the main text.

11) The third sentence of the subsection “The “oxidized” subpopulation is enriched with cells at G1 phase”, seems out of place in the Results.

12) In the Introduction, the authors reference earlier work to state "In addition, early oxidation of the mitochondrial GSH pool preceded subsequent cell death at the population level following exposure to H_2_O_2_ and diatom-derived infochemicals in the model diatom *Phaeodactylum tricornutum*." A key conclusion of their study is that oxidation of the plastid, not the mitochondrium, is a good predictor of cell death. The authors should expand on this in the Discussion.

Reviewer #2:

This is an excellent study on the versatility of the chloroplast oxidation state and subsequent cell fate in a model marine diatom exposed to a range of external oxidative stress. The authors have leveraged a number of powerful and elegant techniques to reveal the heterogeneity of the metabolic stress response at the single-cell level. The study is technically impressive, the graphics are clear and well made, and the manuscript is organized and well written overall. I do have some concerns with the paper, particularly in terms of the discussion of ROS and the relevance of the experimental conditions to the natural environment, which are detailed below.

Hydrogen peroxide is typically in the nanomolar range in seawater (Zinser, 2018. The levels of hydrogen peroxide used in experiments (10's of μM) are therefore much higher than would be expected in most natural marine environments. Can the authors explain in more detail how the chosen hydrogen peroxide concentrations are representative of stress conditions that may be encountered in nature?

The discussion is focused on the paradigm of stress = ROS accumulation = PCD. While this is not incorrect, I do feel that it is oversimplified, and that a broader discussion of the results is more appropriate. For example, ROS were not directly measured, so it cannot be said that changes in E_GSH_ were driven conclusively by ROS dynamics. Accordingly, a discussion of other potential factors besides ROS that could affect E_GSH_ in the chloroplast would be helpful. Indeed, the results showed that chl-roGFP oxidation in the presence of H_2_O_2_ occurred more readily and more permanently in the dark than in the light (Figure 7, Figure 7—figure supplement 2). This is opposite to what would be expected if ROS were involved, because ROS production in the chloroplast should not increase in the dark (it should increase in the light). So it seems that other mechanisms of chl-roGFP oxidation and subsequent mortality are likely at play. Similarly, there is a focus on PCD being the mechanism of mortality. Is PCD the only way for cells to die in monoculture? Or could another mechanism of mortality be involved in association with chloroplast oxidation state? I believe a fuller discussion than currently given would be helpful.

There is growing recognition that ROS play a variety of beneficial signaling and regulatory roles across a range of cell types. The authors allude to this lightly (e.g., "milder ROS cues", "ROS accumulation.… is used to sense the stress and regulate cell fate" (Figure 9 legend). However, the predominant focus of the discussion is on the deleterious potential of biological ROS production. Since endogenous ROS were not measured and linked to any deleterious impacts on cell vitality in this study, I think a more balanced presentation of the potential functions of biological ROS production, including a more explicit acknowledgement of the possible beneficial roles, is appropriate. For instance, endogenous ROS can act as signaling molecules or oxidative stress inducers based on their concentration (Buetler et al., 2004, News Physiol Sci 19:120-123; Saran 2003, Free Radical Research 37(10): 1045). Although the authors did use exogenous H_2_O_2_ concentrations high enough to induce more deleterious effects and the focus of this paper is on oxidative stress, there was not enough mention that lower endogenous ROS levels could be beneficial. This is highly pertinent information to provide in the context of ROS in cell physiology and cellular redox health.

Please describe specifically what kinds of stressors would be expected to induce the chloroplast redox and cell death responses reported in this study. In various instances, the authors refer to "specific" stress conditions or cues (Figure 9 legend), but aside from a general list provided in the Introduction (first paragraph), the exact types of stressors that may be relevant in the context of the findings of this study are not clear.

Reviewer #3:

This manuscript examines how individual cells and populations of a model diatom species respond to stress from hydrogen peroxide exposure. The authors found that a particular dose of stress leads to two subpopulation of cells, which are ultimately indicators of cell fate (survival/death). The authors go on to show that this subpopulation effect is not due to genetic variability in the population, that it is correlated with the cell cycle and physiological state of the population, and that it is dependent on light.

The main contribution of this paper is that it demonstrates phenotypic heterogeneity exists in this particular species response to this level of stress. While the authors do a lot of work to show that these subpopulations are predictive of cell fate, I don't quite get why this is a breakthrough. The physiological response is not really new or surprising to me (i.e. that more heavily damaged cells are more likely to die) and the fact that this only happens at a very particular range of stress exposure (at one concentration, not ~25% lower or higher, and for no more than 100 minutes) implies to me that this subpopulation physiological response occurs rarely in nature.

While I find this paper to be a rigorous examination of this phenomenon (with impressive single cell data!), I do wonder how influential the observation of this phenomenon is and how this paper drives its field forward. I can evaluate this paper based on my own expertise in the field of phenotypic heterogeneity, but have little ability to comment on diatom physiology beyond my knowledge of bacterial physiology. Simply showing that within population heterogeneity exists after a perturbation is no longer a significant contribution from my perspective (see Ackermann, 2015). I would be more inclined to see it as exciting if the authors were able to demonstrate the mechanisms underlying the cause of the heterogeneity or that this heterogeneity has important consequences for the organism or its ecosystem. The authors address this in the Discussion and state that they don't have much evidence for the causes of this heterogeneity or its ecological consequences (Discussion, fifth and last paragraphs). I may be missing something regarding this study’s findings and its relevance to what is known about programmed cell death, diatom physiology, or general marine microbiology. If this is the case I am open to learning more about this, but the paper doesn't frame this point as its central contribution to the field.

Outside of this big picture issue of influence/significance to a general audience, I do find the study to be a strong demonstration of phenotypic heterogeneity which excludes other possible sources of cell-to-cell variability. The methods used and analyses performed are strong. The data is persuasive that phenotypic heterogeneity is happening in response to this particular perturbation.

[Editors’ note: what now follows is the decision letter after the authors submitted for further consideration.]

Thank you for submitting your article "Light-dependent single-cell heterogeneity in the chloroplast redox state regulates cell fate in a marine diatom" for consideration by *eLife*. Your article has been reviewed by two peer reviewers, and the evaluation has been overseen by a Reviewing Editor and Ian Baldwin as the Senior Editor. The reviewers have opted to remain anonymous.

The reviewers have discussed the reviews with one another and the Reviewing Editor has drafted this decision to help you prepare a revised submission.

Summary:

The authors have done an excellent job addressing many of the previous concerns, and the manuscript is significantly improved and will be publishable in *eLife* pending one more round of revision. Upon return, the manuscript will not be sent for re-review; an editorial decision will be made expeditiously.

Essential revisions:

There are a few important yet straightforward editorial changes that should be made prior to publication. The most important of these concern the presentation of the "death threshold" data as well as referencing assertions about environmental H_2_O_2_ concentrations and light intensities.

Reviewer #1:

The authors have substantially improved their manuscript with the changes that have been made. The review process has convinced me that this paper does contribute a more broadly interesting story to readers beyond the scope of the phenotypic heterogeneity story. They have now linked the heterogeneity in chloroplast oxidation state from H_2_O_2_ experiments to a similar phenomenon in a separate environmentally relevant stressor (high light) with new experiments. The clarity and focus of the manuscript is also much improved, particularly in the Introduction/Discussion sections. The discussion of potential mechanisms underlying the light-dependent heterogeneity is also an important improvement. The extra data in Figure 2A also made it much more convincing.

The authors have addressed my previous comments well. I have one final major issue with the way the authors present the OxD% "death threshold" findings in Figure 3 and in the text (subsection “Early oxidation of chloroplast E_GSH_ predicts cell fate at the single-cell level”, last paragraph; Discussion, fifth paragraph). The authors appear to use the same dataset to define a OxD% threshold for predicting live-dead cells, and then test the accuracy of using that threshold for predicting cell fate. This is obviously circular, and a true test of the accuracy needs to use a different dataset/experiment. If the authors did an independent experiment they need to state it. If not, the authors need to modify these statements from the text because they will mislead readers to think that independent validation experiments have been done. Conceptually, I think they have evidence for a death threshold, but they don't have enough data to estimate the true value of that threshold or if the threshold changes with different conditions/stressors. While the exact single cell experiments weren't done to test this for high light stress, it appears that OxD% thresholds leading to death aren't equivalent across at least high-light and H_2_O_2_ (Figures 1, 2, 4, and 8). I would invite the authors to discuss the concept of the "death threshold", but withhold from declaring they have measured a threshold that can accurately predict cell fate (which implies independent validation). I would also suggest they discuss the possibility of the threshold being dependent on the stressor or ecological context.

Reviewer #2:

The authors have significantly revised their manuscript in response to the reviewers' comments. The authors made several changes in response to my major concern regarding the environmental relevance of the study – these include new figures and new data demonstrating that the chloroplast redox response to H_2_O_2_ application is similar in magnitude to other environmentally relevant stimuli, such as high light. I appreciate the additional experiments, but I'm still not sure about the choice of the high light level (~2000 μmol/photons/m^2), which was said to be similar to values measured in surface seawater (subsection “High light induces variability in chloroplast ROS accumulation and subsequent cell survival”, first paragraph), yet this is not general knowledge and I couldn't find any support for this statement in the manuscript. I also accept the argument that H_2_O_2_ concentrations could be higher in microenvironments compared to bulk seawater, but without additional references, I am skeptical that these microenvironments could reach tens of μm H_2_O_2_. Overall, I am partially satisfied that these changes resolve my original concerns. For example, it seems pertinent to discuss what the typical H_2_O_2_ concentrations are within cells under "normal" and oxidative stress conditions, and how these concentrations compared to the levels of H_2_O_2_ used in the experiments (the authors allude to this in their response, but I couldn't find anything on this subject in the manuscript). I accept that not all of the environmental relevance question needs to be addressed in this single manuscript. For example, I agree with the authors that the ecological significance in the marine environment should be addressed in future studies (last line of the Discussion), but I still think there is a bit more that could be easily done to address the environmental relevance of the present work. I would support acceptance of this version of the manuscript, provided that the authors are encouraged to consider the above comments during the proofs stage.

---

## [Author Response]

[Editors’ note: the author responses to the first round of peer review follow.]

Reviewer #1:[…] 1) Overall, the manuscript is well-written, although it was a little repetitive in sections. This is a manuscript packed with data and figures. Removing any repetitiveness would help with the flow. One example is where the authors state:"To further explore the physiological factors that may affect the bi-stable distribution, we investigated the effects of different growth phases on oxidation of chl-roGFP in response to H_2_O_2_. Since aging has been associated with changes in redox states in several model systems^40-42^, we hypothesized that culture growth phase may affect the susceptibility of the population to oxidative stress."This could be simplified to:"To further explore the physiological factors that may affect the bi-stable distribution, we investigated the effects of different growth phases on oxidation of chl-roGFP in response to H_2_O_2_ as aging has been associated with changes in redox states in several model systems^40-42^."

We thank the reviewer for acknowledging the importance of the work and for providing helpful suggestions. After adding the high light section in the Results, we decided to remove the section about growth phase for better flow and readability, and it will be included instead in future research. Therefore, these repetitive parts were removed. In addition, we modified the Discussion and additional parts in the paper to further improve the flow.

2) Introduction, second paragraph – typo: "response" should be replaced with "respond".

Changed accordingly.

3) Materials and methods – growth state is clearly an important parameter for phenotypic variability. The authors should provide a bit more information than "Cultures were kept in exponential phase." How were cultures maintained in exponential phase? This is particularly important for interpreting Figure 6A as the initial slopes appears to differ between the different curves.

Additional information regarding culture growth and maintenance is now provided in the Materials and methods, see subsection “Culture growth”.

4) Materials and methods – There are a number of places within the Materials and methods that include results (see for example, in the subsection “roGFP measurements”) and references to figures that are described in the main text. This can be confusing for the reader and I suggest that the authors remove references to figures and results from the Materials and methods section.

We thank the reviewer, we have removed the results from the revised version accordingly. However, we kept in the Materials and methods section some references to specific figures that are essential for the understanding of the experimental setup.

5) Figure 1A, control. Could the authors comment on the slight shift to a more reduced state at 50 minutes and then the return to the initial "control" state at 282 minutes? Could this reflect cell cycle transitions?

We thank the reviewer for pointing out this interesting observation, however it is beyond the scope of the current work. This observation requires further investigation and validation, and it may be related to cell cycle transitions, diel cycle, or other technical or biological sources. As the manuscript is already packed with data, we feel that adding additional information regarding this observation will divert the reader from the main findings and will not benefit the paper.

6) Figure 1. The authors could consider adding a control distribution to panels 1b-d in a manner similar to what was done with the distributions for fully oxidized and reduced. This would help illustrate any shifts away from/return to the control state over the course of the experiment.

We respectfully disagree, as the control distribution over the entire time course of the experiment is shown in Figure 1A, E and I, and therefore we believe that adding it within the other figures will be redundant. Since the x axis is kept this comparison should be visually clear. In addition, the mean oxidation over time of each subpopulation as well as control cells is shown in Figure 2—figure supplement 3G, further demonstrating shifts in oxidation state.

7) Figure 2B. The schematic panel is not required as part of main text.

We thank the reviewer for the comment, and we therefore rearranged the figure layout: what was previously Figure 2B was moved to the supplementary information and an illustration for the sorting procedure was added (new Figure 2—figure supplement 3), and Figure 2C was removed.

8) Figure 2C legend. I realize this was discussed in the Materials and methods, but in the legend, it is not clear what is meant by "sorted based on positive roGFP fluorescence" given that the sort window is indicated in the 2C panel.

The meaning in “positive roGFP fluorescence” is a clear fluorescence signal of roGFP in both i405 and i488 channels, which is well separated from wild type cells, which possess only auto-fluorescence signal (i.e. cells that don’t express roGFP). This gate is not affected by the degree of oxidation of roGFP. To clarify it, we changed the description in Figure 3B legend as follows: “Control – untreated chl-roGFP positive cells that were sorted regardless of degree of oxidation”. Figure 3C was removed, and instead an illustration for the sorting procedure was added as part of the schematic representation (Figure 3—figure supplement 1). In addition, we further clarified the sorting strategy in the Materials and methods section (see subsections “roGFP measurements” and “Sorting of subpopulations and generation of clonal populations”). Furthermore, in the newly added Figure 8—figure supplement 1 of the high light section the gate for positive roGFP expression is shown.

9) The use of colony forming units to detect survival was initially confusing to me. I realized after reading further that this was done to allow the authors to generate clonal cell lines for subsequent analysis. This could be clarified when first referenced in the main text.

In order to clarify the use of the CFU assay, we added the following sentence: “The CFU assay provides a direct link between chl-roGFP oxidation and the ability of individual cells to proliferate, and in addition enables to generate clonal populations which were also used for downstream analyses (Figure 3—figure supplement 1).”

The use of CFU to measure survival was conducted in order to measure directly the survival of the same cells that their oxidation was measured. In other methods, such as the Sytox staining shown in Figure 3A, there is a correlation between the fraction of cells that were oxidized and the fraction of cells that were dead but these are not the same cells. In addition, Sytox staining will only detect cells that are dead or dying at the time of measurement, and not cells that were already completely disintegrated or cells that will die at a later stage. Furthermore, the Sytox measurement of sorted subpopulations (shown in Figure 3—figure supplement 2) requires a larger quantity of sorted cells for the analysis and prolongs sorting times. However, the CFU assay measures survival in a different manner, by the ability of the cell to proliferate, which may be even more biologically relevant. In that assay, the “time of death” is not relevant, and also cells with permanent growth arrest will appear as non-viable, providing a direct method to investigate the subsequent fate of individual cells. In addition, the same assay was also used for generating clonal populations.

10) Figure 3H (the diagram) should be moved to the supplement rather than included as part of the main figure. In fact, this panel is not currently referenced in the main text.

As suggested by the reviewer the figure was moved to the supplementary accordingly (new Figure 4—figure supplement 1B).

11) The third sentence of the subsection “The “oxidized” subpopulation is enriched with cells at G1 phase”, seems out of place in the Results.

This sentence was removed, and the subject is addressed in the Discussion (sixth paragraph).

12) In the Introduction, the authors reference earlier work to state "In addition, early oxidation of the mitochondrial GSH pool preceded subsequent cell death at the population level following exposure to H2O2 and diatom-derived infochemicals in the model diatom Phaeodactylum tricornutum." A key conclusion of their study is that oxidation of the plastid, not the mitochondrium, is a good predictor of cell death. The authors should expand on this in the Discussion.

We now elaborate on the subject in the Discussion (fourth paragraph).

Reviewer #2:[…] I do have some concerns with the paper, particularly in terms of the discussion of ROS and the relevance of the experimental conditions to the natural environment, which are detailed below.Hydrogen peroxide is typically in the nanomolar range in seawater (Zinser, 2018). The levels of hydrogen peroxide used in experiments (10's of μM) are therefore much higher than would be expected in most natural marine environments. Can the authors explain in more detail how the chosen hydrogen peroxide concentrations are representative of stress conditions that may be encountered in nature?

We thank the reviewer for raising this important point. We added an explanation regarding the H_2_O_2_ concentrations measured in the environment accordingly (Introduction, third paragraph). To clarify the reason for choosing the specific H_2_O_2_ concentrations used in the study we added Figure 1 and the following explanation: “Next, we examined the response of cells to oxidative stress by treatment with H_2_O_2_ at concentrations that led to oxidation patterns similar to other environmental stressors (Figure 1) and that also led to death of part of the population^16^”. Importantly, the H_2_O_2_ concentrations applied in our study (10’s µM) are three orders of magnitude lower (at least) compared to concentrations that are commonly used in numerous papers in plants, green algae, yeast and other protists (10’s mM).

We agree with the reviewer that H_2_O_2_ concentrations measured in averaged bulk seawater are much lower than the concentrations used in this study. However, these measurements are not derived necessary from high dense algal blooms and do not accurately depict the actual conditions that a single-celled phytoplankton will actually experience. The marine environment is not homogenous, but rather patchy in time and space^8–10^. The cell densities experienced by microorganisms in their microenvironment can be significantly higher due to local high densities in marine snow, biofilm, aggregates, and at the peak of coastal blooms, when cell densities can reach up to 10^5^ cells per mL of seawater^8,9^. Intracellular ROS concretions produced by the cells themselves or by bystanders cells could lead to higher intracellular concentrations than measured in bulk seawater in the field, and can generate a gradient of a few orders of magnitude across the phycosphere^8–10^. As was shown in previous studies, diverse environmental stressors lead to similar oxidation patterns (assessed in vivo by roGFP) as observed in response to H_2_O_2_ (see the added new Figure 1)^1,2^, or may lead to H_2_O_2_ production^3–7^. More specifically, high light, nitrogen limitation, and cyanogen bromide (a toxic infochemical produced by some diatoms, applied in environmentally relevant concentrations) were shown to cause chl-roGFP oxidation in *P. tricornutum*, similarly to the response to H_2_O_2_ in concentrations used in this study. To further address the issue of environmental relevance we examined the effects of high light, as shown in the newly added section in the Results (subsection “High light induces variability in chloroplast ROS accumulation and subsequent cell survival”). High light can lead to H_2_O_2_ accumulation in the chloroplast through photoreduction of O_2_ to superoxide, which in turn is rapidly converted to H_2_O_2_ by superoxide dismutase (SOD)^4–6^. Indeed, following exposure to high light a distinct oxidized subpopulation emerged (Figure 7A-B, Figure 8—figure supplement 1A-F), similarly to the response to external addition of H_2_O_2_. The survival of the oxidized subpopulation was initially high, but then gradually decreased over time of exposure to high light (Figure 7D). These findings demonstrate a similar pre-commitment phase to that observed in response to H_2_O_2_ addition, during which cells were still able to recover upon change of conditions (i.e. by sorting into fresh media, Figure 7D). Taken together with the increased sensitivity to oxidized stress observed under dark conditions, these results demonstrate that light regime plays an important role in determining cell fate and sensitivity to additional stressors in marine diatoms. Light fluctuations are frequent in marine environments due to mixing, shading, diel cycle and tide (in coastal and intertidal regions), and therefore these results are highly relevant to environmental scenarios.

The discussion is focused on the paradigm of stress = ROS accumulation = PCD. While this is not incorrect, I do feel that it is oversimplified, and that a broader discussion of the results is more appropriate. For example, ROS were not directly measured, so it cannot be said that changes in E_GSH_ were driven conclusively by ROS dynamics. Accordingly, a discussion of other potential factors besides ROS that could affect E_GSH_ in the chloroplast would be helpful. Indeed, the results showed that chl-roGFP oxidation in the presence of H_2_O_2_ occurred more readily and more permanently in the dark than in the light (Figure 7, Figure 7—figure supplement 2). This is opposite to what would be expected if ROS were involved, because ROS production in the chloroplast should not increase in the dark (it should increase in the light). So it seems that other mechanisms of chl-roGFP oxidation and subsequent mortality are likely at play.

Indeed, the oxidation observed in chl-roGFP in the dark is surprising, and the sources for ROS production in the dark are yet to be further investigated. One possible explanation could be that the recycling of antioxidants is impaired in the dark, as light-generated NADPH is used for the recycling and biosynthesis of GSH and other antioxidants. The abrupt transition to the dark during daytime may have caused a perturbation in the redox balance through the inability to recycle GSSG to GSH, leading to oxidation of chl-roGFP. This could also explain the increase in sensitivity when cells are under dark, which was also observed in another diatom species^11^ (see Discussion, seventh paragraph).

As detailed in previous works, the redox sensor roGFP is coupled to the redox state of GSH through the activity of glutaredoxins (GRX), for which roGFP serves as an artificial target that is mimicking native proteins with similar redox potential (for a comprehensive review on roGFP sensors see

Meyer and Dick, 2010)^12,13^. Perturbations in the balance between GSHGSSG are due to changes in the redox state in their subcellular microenvironment, which are mainly driven by ROS dynamics. Therefore, roGFP measures the biochemical response of the cell to ROS through its coupling to the E_GSH_ balance. Changes in the size of the GSH and GSSG pools without changing the balance between them (i.e. without changes in E_GSH_) should not affect roGFP oxidation. Therefore, we are not sure to which additional sources for roGFP oxidation the reviewer is referring. The great advantage of using the roGFP as an in vivo sensor over any ROS chemical fluorescent stains is that it can be reversibly oxidized and reduced, and thus is sensitive to the actual fluctuations of the redox state within a given microenvironment. An additional big difference is that the roGFP is a genetically encoded probe, and thus can be targeted to measure ROS in a specific organelle, in contrast to a chemical stain that accumulates by diffusion in a non-specific manner, and may also accumulate in different cells to a different degree due to permeability differences. The roGFP is a ratiometric probe, and its oxidation measurements are therefore less affected by bleaching or variability in expression levels.

Regarding the role of ROS in simply mediating PCD, we respectfully disagree with the reviewer. The whole ideology of our body of works in the past 5 years is around how specificity in redox singling can regulate not only PCD, but also intricate signaling cascades that lead to cell survival under diverse stress conditions^1–3,14,15^. Over the years, we revealed how redox changes due to environmental stresses can generate highly specific redox patterns in specific organelles and to activate a redox-sensitive proteome network that regulates metabolic fluxes and cell fate decisions. In our current manuscript, we take a step further to show that under the same given stress of high light or H_2_O_2_, individual cells within the same isogenic population can display distinct cell fates, of PCD and also survival. Furthermore, cells of the “reduced” subpopulation are shown to exhibit mild chl-roGFP oxidation in response to H_2_O_2_ treatment (Figures1C, 3F and Figure 2—figure supplement 3G), and are also shown to divide and survive under exposure to oxidative stress (Figures 2B, 3F, and Figure 4—video 1). Please see additional information below.

Similarly, there is a focus on PCD being the mechanism of mortality. Is PCD the only way for cells to die in monoculture? Or could another mechanism of mortality be involved in association with chloroplast oxidation state? I believe a fuller discussion than currently given would be helpful.

To the best of our knowledge, the main mode of death in response to H_2_O_2_ in concentrations that were used in this manuscript has clear PCD markers. Our previously published results (as mentioned in the third paragraph of the Introduction and in the fifth paragraph of the Discussion) demonstrated that H_2_O_2_ led to the induction of cell death with PCD characteristics in *P. tricornutum*, consisting of a cascade of events of DNA fragmentation (starting in at least part of the population within 4 h of treatment, measured by TUNEL assay), followed by phosphatidylserine externalization (starting 8 h post treatment, assayed by Annexin V), and eventually irreversible membrane permeabilization (positive Sytox staining 24 h post treatment)^1^. Lastly, we have recently characterized the first PCD molecular component from *P. tricornutum*, metacaspase 5, which is regulated by H_2_O_2_ and infochemicals produced by diatoms^16^. Nevertheless, we agree with the reviewer that there are other modes of death in monoculture, such as direct cell lysis, necrosis and autophagic cell death etc. We clarified the use of the PCD term for describing the mode of death in our system in the Introduction and the Discussion (see the aforementioned paragraphs), however we feel that further mechanistic discussion of alternative modes of death in response to other stressors is out of the scope of this paper.

There is growing recognition that ROS play a variety of beneficial signaling and regulatory roles across a range of cell types. The authors allude to this lightly (e.g., "milder ROS cues", "ROS accumulation.… is used to sense the stress and regulate cell fate" (Figure 8 legend). However, the predominant focus of the discussion is on the deleterious potential of biological ROS production. Since endogenous ROS were not measured and linked to any deleterious impacts on cell vitality in this study, I think a more balanced presentation of the potential functions of biological ROS production, including a more explicit acknowledgement of the possible beneficial roles, is appropriate. For instance, endogenous ROS can act as signaling molecules or oxidative stress inducers based on their concentration (Buetler et al. 2004, News Physiol Sci 19:120-123; Saran 2003, Free Radical Research 37(10): 1045). Although the authors did use exogenous H2O2 concentrations high enough to induce more deleterious effects and the focus of this paper is on oxidative stress, there was not enough mention that lower endogenous ROS levels could be beneficial. This is highly pertinent information to provide in the context of ROS in cell physiology and cellular redox health.

In this work and in our previous publications^1–3,14,15^ we strongly emphasize the role of ROS as signaling molecules used in sensing stress cues and mediating the response to these cues. In our work, we propose that redoxbased responses to stress serve as a mechanism for a rapid and reversible response to variable conditions, and act as a powerful surveillance system to accurately assess the chemo-physical gradients in the cell’s microenvironment, enabling discrimination between multiple environmental cues (for example see the Introduction, Discussion, second and third paragraphs, and the added Figure 1). In addition, in the Introduction we specifically emphasize the role of H_2_O_2_ in signaling (third paragraph). The results presented in this paper further support a signaling role for ROS, which mediate cell fate determination to either survival or death. For example, we show that also in the “reduced” subpopulation chl-roGFP underwent rapid oxidation in response to H_2_O_2_ treatment, however to much lower values than the “oxidized” subpopulation (30-65% OxD compared to ~100% OxD), and afterwards recovered from the stress (Figures 2B-D, 4F, Figure 2—figure supplement 3G, and Figure 4—video 1). Moreover, cells of the “reduced” subpopulation were also able to proliferate despite the stress conditions (Figure 4—video 1). In addition, the E_GSH_ oxidation in the chloroplast is proposed as a sensing mechanism, as chl-roGFP oxidation preceded the cell fate determination in the “oxidized” subpopulation (Discussion, third paragraph). Furthermore, we demonstrate that upon transition of *P. tricornutum* cells to the dark, chl-roGFP oxidized immediately and then gradually reduced over time, and this response was not accompanied with increased death, and was specific to the chloroplast as it was not observed in the nucleus (Figure 7A and E, Figure 7—figure supplement 2, subsection “The bimodal chloroplast redox response is light dependent”, first paragraph). To further clarify the subject we modified the Discussion section.

Please describe specifically what kinds of stressors would be expected to induce the chloroplast redox and cell death responses reported in this study. In various instances, the authors refer to "specific" stress conditions or cues (Figure 8 legend), but aside from a general list provided in the Introduction (first paragraph), the exact types of stressors that may be relevant in the context of the findings of this study are not clear.

As was shown in our previous studies, diverse environmental stressors lead to similar oxidation patterns as observed in response to H_2_O_2_ (see the newly added Figure 1)^1,2^, or may lead to H_2_O_2_ production^3–7^. More specifically, high light, nitrogen limitation, cyanogen bromide (a toxic infochemical produced by some diatoms) were shown to cause roGFP oxidation in the chloroplast in *P. tricornutum*. To further address this important issue of environmental relevance we examined the effects of high light, an important stressor for photosynthetic microbes in the aquatic ecosystems, as shown in Figures 8, Figure 8—figure supplement 1 and Figure 8—figure supplement 2, and in the adequate section in the Results (subsection “High light induces variability in chloroplast ROS accumulation and subsequent cell survival”). High light can lead to H_2_O_2_ accumulation in the chloroplast through photoreduction of O_2_ to superoxide, which in turn is rapidly converted to H_2_O_2_ by superoxide dismutase (SOD)^4–6^. Indeed, following exposure to high light a distinct oxidized subpopulation emerged, similarly to the response to external addition of H_2_O_2_ (Figure 7A-B and Figure 7—figure supplement 2A-F). The survival of the oxidized subpopulation gradually decreased over time of exposure to high light, exhibiting a similar pre-commitment phase to that observed in response to H_2_O_2_ addition, during which cells were still able to recover upon change of conditions (i.e. by sorting into fresh media, Figure 7D). Taken together with the increased sensitivity to oxidized stress observed under dark conditions, these results demonstrate that light regime plays an important role in determining cell fate and sensitivity to additional stressors in marine diatoms. Light fluctuations are frequent in marine environments due to mixing, shading, diel cycle and tidal oscillations (in coastal and intertidal regions), and therefore these results are highly relevant to environmental scenarios. We modified the Discussion accordingly and added in Figure 8 high light and infochemicals as possible stressors.

Reviewer #3:[…] While I find this paper to be a rigorous examination of this phenomenon (with impressive single cell data!), I do wonder how influential the observation of this phenomenon is and how this paper drives its field forward. I can evaluate this paper based on my own expertise in the field of phenotypic heterogeneity, but have little ability to comment on diatom physiology beyond my knowledge of bacterial physiology. Simply showing that within population heterogeneity exists after a perturbation is no longer a significant contribution from my perspective (see Ackermann, 2015). I would be more inclined to see it as exciting if the authors were able to demonstrate the mechanisms underlying the cause of the heterogeneity or that this heterogeneity has important consequences for the organism or its ecosystem. The authors address this in the Discussion and state that they don't have much evidence for the causes of this heterogeneity or its ecological consequences (Discussion, fifth and last paragraphs). I may be missing something regarding this study’s findings and its relevance to what is known about programmed cell death, diatom physiology, or general marine microbiology. If this is the case I am open to learning more about this, but the paper doesn't frame this point as its central contribution to the field.

We thank the reviewer for acknowledging the rigorosity of the work, but we respectfully disagree with the reviewer summarizing the paper as simply another paper describing phenotypic heterogeneity. We feel that the main problem is the knowledge gap between the well-established bacterial community, which over the last 2 decades has demonstrated the huge importance of phenotypic heterogeneity among bacterial model systems, and the very young field of cell biology and genetics of marine phytoplankton. The subject of phenotypic variability has been studied extensively in bacterial, yeast and mammalian model systems^17–19^, mainly in model systems with little ecological relevance to the marine environment. In contrast, very little is known or reported for marine microbes in general, and specifically in marine phytoplankton. Especially, there was never specific effort to track single cell response in high resolution in order to better understand how unicellular algae sense, respond and communicate stress and chemical cues. We believe that our paper is a first step towards an important discussion about the relevance of phenotypic heterogeneity in aquatic ecosystems, and a fascinating research question to specifically explore in the context of phytoplankton bloom dynamics. Importantly, to date little is known about the genetic programs and signal transduction pathways that mediate perception of abiotic and biotic stresses that phytoplankton are typically being subjected to during natural bloom succession. We propose that our approach will enable the discovery of essential genes that mediate acclimation strategies, and will include genes involved in activation of PCD and in cellular defense and recovery. Current understanding of phytoplankton responses to environmental stressors is based primarily on bulk analyses, neglecting cell-to-cell heterogeneity. Here, we established a novel system for studying redox-based heterogeneity in diatoms, and it can be used in the future to study other ecologically relevant microorganisms. We believe that our manuscripts greatly contributes to the field, as shown by the major important novel findings:

- This is the first study to track redox-based heterogeneity in the chloroplast in response to oxidative stress and high light, leading to differential survival within isogenic populations.

- This is the first report to assign heterogeneity to light dependent metabolic processes.

- We demonstrate that chloroplast E_GSH_ is involved in sensing stress cues and mediating cell fate determination in algae. Specifically, we show a link between ROS accumulation in the chloroplast and subsequent cell death. The role of the chloroplast in mediating activation and regulation of PCD is still an open question in photosynthetic organisms including plants^20,21^, in contrast to the well-established mammalian literature on the pivotal role of the mitochondria in executing the PCD signaling cascade. Our results shed light on the role of light-dependent reactions in the chloroplast on regulating cell fate in diatoms.

- We provide insights into the time-line of events in the response to oxidative stress and the PCD cascade in diatoms. Chloroplast E_GSH_ oxidation changes within minutes and precedes cell fate determination, exposing the “pre-commitment” phase that ranges 30-60 min in most cells.

- In order to add an important ecological context to the main findings of the manuscript, regarding the bi-stable response to oxidative stress in marine diatoms, we examined the response of the cells to high light. We demonstrate that high light, a key environmental stress factor in the ocean, controls diatoms’ stress physiology. Specifically, we show that high light can induce the emergence of distinct subpopulations, differing in ROS accumulation in the chloroplast and in their subsequent survival. In addition, transition to the dark greatly enhanced the sensitivity to oxidative stress. Together, these results may have great ecological implications, as phytoplankton experience frequent light intensity fluctuations due to mixing, the diel cycle, shading, and tide (in intertidal and coastal regions). Therefore, the time of day and position within the water column may greatly affect diatoms’ biotic interactions with other organisms in the marine environment and their ability to cope with additional stressors as nutrient limitation, temperature gradients and more.

- It is important to emphasize that although there are numerous studies examining heterogeneity, many rely on gene expression as their readout. In the current work we track an important metabolic readout of redox changes in vivo in an organelle specific manner, similar to recent efforts to probe NADPH, glucose etc. Our unique readout is strongly associated with ROS metabolism and thus gives an important view on the cellular stress physiology at the single cell and organelle levels.

As was shown in previous studies, diverse environmental stressors lead to similar oxidation patterns as observed in response to H_2_O_2_ (see the added Figure 1)^1,2^, or may lead to H_2_O_2_ production^3–7^. More specifically, high light, nitrogen limitation and cyanogen bromide (a toxic infochemical produced by some diatoms) were shown to cause roGFP oxidation in the chloroplast in *P. tricornutum* (Figure 1). To further address the issue of environmental relevance we examined the effects of high light, as shown in the added section in the Results (subsection “High light induces variability in chloroplast ROS accumulation and subsequent cell survival”). High light can lead to H_2_O_2_ accumulation in the chloroplast through photoreduction of O_2_ to superoxide, which in turn is rapidly converted to H_2_O_2_ by superoxide dismutase (SOD)^4–6^. Indeed, following exposure to high light a distinct “oxidized” subpopulation emerged, similarly to the response to external addition of H_2_O_2_ (Figure 7A-B). The survival of the “oxidized” subpopulation gradually decreased over time of exposure to high light, exhibiting a similar pre-commitment phase to that observed in response to H_2_O_2_ addition, during which cells were still able to recover upon change of conditions (i.e. by sorting into fresh media, Figure 7D). As explained above, these results are highly relevant to environmental scenarios.

We fully agree with the reviewer that the next critical step must be to unravel the underlying molecular mechanism that will regulate cell fate decisions in marine diatoms. We are currently investigating this challenging subject using the exciting system developed here as the fundamental experimental setup. We therefore aim to develop single-cell RNAseq protocols adapted to marine phytoplankton, which will enable in the near future to expose the transctriptome makeup that defines the different subpopulations with contrasting cell fates. Nevertheless, our data is still very preliminary and beyond the scope of the current manuscript, which is already packed with data.

Outside of this big picture issue of influence/significance to a general audience, I do find the study to be a strong demonstration of phenotypic heterogeneity which excludes other possible sources of cell-to-cell variability. The methods used and analyses performed are strong. The data is persuasive that phenotypic heterogeneity is happening in response to this particular perturbation.

We greatly appreciate the reviewer encouraging feedback on the quality of the work.

1) Graff van Creveld, S., Rosenwasser, S., Schatz, D., Koren, I. and Vardi, A. Early perturbation in mitochondria redox homeostasis in response to environmental stress predicts cell fate in diatoms. ISME J. 9, 385–395 (2015).

2) Rosenwasser, S. et al. Mapping the diatom redox-sensitive proteome provides insight into response to nitrogen stress in the marine environment.

Proc. Natl. Acad. Sci. U. S. A. 111, 2740–5 (2014).

3) Sheyn, U., Rosenwasser, S., Ben-Dor, S., Porat, Z. and Vardi, A. Modulation of host ROS metabolism is essential for viral infection of a bloom-forming coccolithophore in the ocean. ISME J. 10, 1–13 (2016).

4) Asada, K. Production and scavenging of reactive oxygen species in chloroplasts and their functions. Plant Physiol. 141, 391–6 (2006).

5) Waring, J., Klenell, M., Bechtold, U., Underwood, G. J. C. and Baker, N. R.

Light-induced responses of oxygen photoreduction, reactive oxygen species production and scavenging in two diatom species. J. Phycol. 46, 1206–1217 (2010).

6) Exposito-Rodriguez, M., Laissue, P. P., Yvon-Durocher, G., Smirnoff, N. and Mullineaux, P. M. Photosynthesis-dependent H_2_O_2_ transfer from chloroplasts to nuclei provides a high-light signalling mechanism. Nat. Commun. 8, 49 (2017).

7) Vardi, A. et al. Programmed cell death of the dinoflagellate Peridinium gatunense is mediated by CO2 limitation and oxidative stress. Curr. Biol. 9, 1061–1064 (1999).

8) Seymour, J. R., Amin, S. A., Raina, J.-B. and Stocker, R. Zooming in on the phycosphere: the ecological interface for phytoplankton–bacteria relationships. Nat. Microbiol. 2, 17065 (2017).

9) Stocker, R. Marine Microbes See a Sea of Gradients. Science 338, 628– 633 (2012).

10) Chrachri, A., Hopkinson, B. M., Flynn, K., Brownlee, C. and Wheeler, G. L. Dynamic changes in carbonate chemistry in the microenvironment around single marine phytoplankton cells. Nat. Commun. 9, 74 (2018).

11) Volpert, A., Graff van Creveld, S., Rosenwasser, S. and Vardi, A. Diurnal fluctuations in chloroplast GSH redox state regulate susceptibility to oxidative stress and cell fate in a bloom-forming diatom. J. Phycol. 341, 329–341 (2018).

12) Meyer, A. J. et al. Redox-sensitive GFP in *Arabidopsis thaliana* is a quantitative biosensor for the redox potential of the cellular glutathione redox buffer. Plant J. 52, 973–986 (2007).

13) Meyer, A. J. and Dick, T. P. Fluorescent protein-based redox probes. Antioxid Redox Signal 13, 621–650 (2010).

14) Creveld, S. G. Van, Ben-dor, S., Mizrachi, A., Alcolombri, U. and Hopes, A. A redox-regulated type III metacaspase controls cell death in a marine diatom. (2018). doi:https://doi.org/10.1101/444109

15) Graff van Creveld, S., Rosenwasser, S., Levin, Y. and Vardi, A. Chronic Iron Limitation Confers Transient Resistance to Oxidative Stress in Marine Diatoms. Plant Physiol. 172, 968–979 (2016).

16) Creveld, S. G. Van, Ben-dor, S., Mizrachi, A., Alcolombri, U. and Hopes, A. A redox-regulated type III metacaspase controls cell death in a marine diatom. (2018). doi:https://doi.org/10.1101/444109

17) Balaban, N. Q. Persistence: Mechanisms for triggering and enhancing phenotypic variability. Curr. Opin. Genet. Dev. 21, 768–775 (2011).

18) Raj, A. and van Oudenaarden, A. Nature, Nurture, or Chance: Stochastic Gene Expression and Its Consequences. Cell 135, 216–226 (2008).

19) Ackermann, M. A functional perspective on phenotypic heterogeneity in microorganisms. Nat. Rev. Microbiol. 13, 497–508 (2015).

20) Van Aken, O. and Van Breusegem, F. Licensed to Kill: Mitochondria, Chloroplasts, and Cell Death. Trends Plant Sci. 20, 754–766 (2015).

21) Lam, E., Kato, N. and Lawton, M. Programmed cell death, mitochondria and the plant hypersensitive response. Nature 411, 848–853 (2001).

[Editors' note: the author responses to the re-review follow.]

Essential revisions:There are a few important yet straightforward editorial changes that should be made prior to publication. The most important of these concern the presentation of the "death threshold" data as well as referencing assertions about environmental H2O2 concentrations and light intensities. Reviewer #1: […] The authors have addressed my previous comments well. I have one final major issue with the way the authors present the OxD% "death threshold" findings in Figure 3 and in the text (subsection “Early oxidation of chloroplast EGSH predicts cell fate at the single-cell level”, last paragraph; Discussion, fifth paragraph). The authors appear to use the same dataset to define a OxD% threshold for predicting live-dead cells, and then test the accuracy of using that threshold for predicting cell fate. This is obviously circular, and a true test of the accuracy needs to use a different dataset/experiment. If the authors did an independent experiment they need to state it. If not, the authors need to modify these statements from the text because they will mislead readers to think that independent validation experiments have been done. Conceptually, I think they have evidence for a death threshold, but they don't have enough data to estimate the true value of that threshold or if the threshold changes with different conditions/stressors. While the exact single cell experiments weren't done to test this for high light stress, it appears that OxD% thresholds leading to death aren't equivalent across at least high-light and H2O2 (Figures 1, Figure 2, 4, and 8). I would invite the authors to discuss the concept of the "death threshold", but withhold from declaring they have measured a threshold that can accurately predict cell fate (which implies independent validation). I would also suggest they discuss the possibility of the threshold being dependent on the stressor or ecological context.

We thank the reviewer for acknowledging the importance of the work and for appreciating the substantial improvement of the paper. We also thank the reviewer for the constructive feedback, and agree with his comments regarding the “death threshold”, as the same dataset was used for generating the threshold and for testing it. According to the reviewer suggestions, we now discuss the concept of a possible “death threshold”, which may be exposed by the chl-roGFP level of oxidation, and discuss the possibility that additional factors may influence this threshold (see subsection “Early oxidation of chloroplast E_GSH_ is linked to cell fate determination at the single-cell level”, last paragraph and Discussion, fifth paragraph). Accordingly, we removed any declarations regarding actual measurements of a threshold and its application in cell fate predictions.

Reviewer #2:The authors have significantly revised their manuscript in response to the reviewers' comments. The authors made several changes in response to my major concern regarding the environmental relevance of the study – these include new figures and new data demonstrating that the chloroplast redox response to H_2_O_2_ application is similar in magnitude to other environmentally relevant stimuli, such as high light. I appreciate the additional experiments, but I'm still not sure about the choice of the high light level (~2000 μmol/photons/m^2), which was said to be similar to values measured in surface seawater (subsection “High light induces variability in chloroplast ROS accumulation and subsequent cell survival”, first paragraph), yet this is not general knowledge and I couldn't find any support for this statement in the manuscript. I also accept the argument that H_2_O_2_ concentrations could be higher in microenvironments compared to bulk seawater, but without additional references, I am skeptical that these microenvironments could reach tens of μm H_2_O_2_. Overall, I am partially satisfied that these changes resolve my original concerns. For example, it seems pertinent to discuss what the typical H_2_O_2_ concentrations are within cells under "normal" and oxidative stress conditions, and how these concentrations compared to the levels of H_2_O_2_ used in the experiments (the authors allude to this in their response, but I couldn't find anything on this subject in the manuscript). I accept that not all of the environmental relevance question needs to be addressed in this single manuscript. For example, I agree with the authors that the ecological significance in the marine environment should be addressed in future studies (last line of the Discussion), but I still think there is a bit more that could be easily done to address the environmental relevance of the present work. I would support acceptance of this version of the manuscript, provided that the authors are encouraged to consider the above comments during the proofs stage.

We thank the reviewer for the positive feedback and for his comments, and we now further elaborated the explanation for the chosen high light (HL) conditions and H_2_O_2_ concentrations and their environmental relevance, as well as providing additional key references (see Introduction, subsection “High light induces variability in chloroplast ROS accumulation and subsequent cell survival”). The HL conditions used in this study (2,000 µmol photons m^-2^ s^-1^) are equivalent to full sunlight conditions in nature, and are likely experienced by phytoplankton in surface seawater (Long, Humphries, and Falkowski 1994). For example, in central subtropical oceans cells may be trapped at the surface layer for at least part of the day due to the diurnal thermocline, which reaches 2-5 m depth during the day (Long et al., 1994). In addition, the same HL conditions were used in a previous study of HL acclimation in *P. tricornutum* (Lepetit et al., 2013). Milder HL conditions may also elicit a similar response perhaps with a milder magnitude, that may lead to a smaller fraction of oxidized cells compared to the HL conditions used in this study (for example see the chl-roGFP oxidation in response to 700 µmol photons m^-2^ s^-1^ in bulk measurements in Figure 1). However, we focused on understanding whether high light could generate an equivalent oxidation level in the chloroplast as exogenous application of H_2_O_2_. Therefore, we chose to use harsher HL conditions that also occur in natural environments, and did not examine in this study the full range of light intensities, which can be further investigated in detail in future work.

Regarding the concentrations of exogenous application of H_2_O_2_ used in this study; it is important to note that our analysis revealed that during diverse environmental stressors, such as toxic infochemicals, different light intensities and nutrient limitation, the changes in chloroplast targeted roGFP degree of oxidation are comparable to the effect of H_2_O_2_ concentrations that were applied exogenously in this study (see Figures 1 and 8). This data suggests that the applied H_2_O_2_ concentrations are relevant to physiological stress conditions and have similar effects on the chloroplast E_GSH_ pool. To demonstrate in a more accessible manner that the response to H_2_O_2_ as applied in this study is comparable with other environmental stressors, we now present this as Figure 1.

To the best of our knowledge, until now there are no direct in vivo measurements of absolute intracellular H_2_O_2_ quantification, in marine microbes (including diatoms) or in their microenvironment. This knowledge gap mainly derives from the lack of tools required for such H_2_O_2_ quantification, and methods as genetically encoded H_2_O_2_ reporters were only developed in recent years and to the best of our knowledge were not yet implemented in algae. Current knowledge is based mainly on bulk measurements, which mask local and temporal fluctuations, or based on relative or indirect measurements typically assessed by chemical fluorescent probes that do not provide absolute in vivo intracellular quantification and without the high spatial resolution that is required to allow measurements of the cell’s microenvironment.

The reviewer refers to 10’s of nM that were measured in the marine environment in a non-bloom, steady state, extracellular bulk measurements. We would like to stress the point that these measurements are completely different from the intracellular, organelle-specific H_2_O_2_ net production after taking into account the cellular antioxidant capacity. In addition, according to Seymour et al., the local concentrations of different compounds in the phycosphere of an individual cell could be several orders of magnitude higher than in bulk seawater (Seymour et al., 2017). Taken together, it is likely that algae cells experience H_2_O_2_ concentrations in the range of µM at least under certain stress conditions, either due to intracellular production (for example in the chloroplast under high light conditions), or due to local or temporal production by neighbor cells. For example, some harmful bloom-forming algae species were shown to produce extracellular H_2_O_2_ and superoxide (which can be rapidly converted to H_2_O_2_) in rates of tens and hundreds of fmol cell^−1^ h^−1^ (Diaz et al., 2018). As stated in the manuscript, the environmental relevance of the bi-stable phenotype is yet to be further investigate in future works, and we aim to address additional environmental stress conditions as well as the possible ecological implications in our future research.